# Physically based approaches incorporating evaporation for early warning predictions of rainfall-induced landslides

Alfredo Reder[1,2], Guido Rianna[2], Luca Pagano[1]

[1]Department of Civil, Architectural and Environmental Engineering, University of Naples Federico II, Naples, 80125, Italy
[2]Regional Models and geo-Hydrological Impacts Division, CMCC Foundation, Capua, 81043, Italy

*Correspondence to*: Guido Rianna (guido.rianna@cmcc.it)

**Abstract.** In the field of rainfall-induced landslides on sloping covers, models for early warning predictions require an adequate trade-off between two aspects: prediction accuracy and timeliness. When a cover's initial hydrological state is a determining factor in triggering landslides, taking evaporative losses into account (or not) could significantly affect both aspects. This study evaluates the performance of three physically based predictive models, converting precipitation and evaporative fluxes into hydrological variables useful in assessing slope safety conditions. Two of the models incorporate evaporation, with one representing evaporation as both a boundary and internal phenomenon, and the other only a boundary phenomenon. The third model totally disregards evaporation. Model performances are assessed by analysing a well-documented case study involving a two-meter thick sloping volcanic cover. The large amount of monitoring data collected for the soil involved in the case study, reconstituted in a suitably equipped lysimeter, makes it possible to propose procedures for calibrating and validating the parameters of the models. All predictions indicate a hydrological singularity at the landslide time (alarm). Comparison of the models' predictions also indicates that the greater the complexity and completeness of the model, the lower the number of predicted hydrological singularities when no landslides occur (false alarms).

## 1 Introduction

In Italy, many sloping deposits of sand or silty sand, constituting covers not exceeding few meters, experience unsaturated conditions throughout the hydrological year. Such state condition permits them to be stable also for slope angles exceeding friction angles (Pagano et al., 2008) thanks to additional strength provided by suction (usually, they are characterized by low/null true cohesion values). The sequence of rainfall events occurring over the wet season induces a general reduction in suction levels, increasing the cover's susceptibility to an exceptional rainfall event. On the other hand, evaporative fluxes reduce susceptibility to sliding by increasing suction levels. The antecedent period, during which the contrast between rainfall and evaporation affects suction levels, may last weeks or months depending on the hydraulic properties of the soils involved and the climate regime of the area (Rahimi et al., 2011; Rahardjo et al., 2001).

The 2005 Nocera Inferiore landslide [hereinafter "2005NIL"] was interpreted (Pagano et al., 2010) by merely referring to precipitation recorded by a meteorological station placed near the landslide area (Fig. 1). Richards' equation (1931) in 1-D

flow conditions was adopted to convert hourly precipitation records into the evolution of soil suction at various depths. This simple approach highlighted the crucial role of antecedent rainfall (945 mm of rainfall over 4.5 months), which had reduced soil suction to very low values before the occurrence of the major event (143 mm of rainfall over 16 hours). Numerical analyses indicate that suction vanishing throughout the entire cover depth can induce the attainment of slope failure conditions. Virtual scenarios built with modified antecedent rainfalls were analysed, and they indicated that the phenomenon would have not occurred if the antecedent periods had been drier. The crucial factor affecting soil suction at triggering time was the weather conditions over the previous four months.

A meteorological window of such long influence implies that it would not be reasonable to neglect evaporative fluxes, as their persistency could result in significant drying processes even during the cold season, when evaporation is at its lowest (about 1-2 mm/day in winter). Rianna et al. (2014a) measured infiltrating precipitation and actual evaporation induced by the actual weather conditions on a layer placed in a lysimeter, made using the same soil that was involved in the 2005NIL. Monitoring showed that over a hydrological year the amount of actual evaporation occurs to the same order of magnitude as that of infiltrating precipitation (hundreds of millimeters).

Increasing efforts are being made to develop early warning systems to mitigate the risks of rainfall-induced landslides. Their success strongly depends on the performance of the predictive models they implement in terms of timing and accuracy of prediction. In tens of centimetres thin and/or coarse-grained soil covers, prediction accuracy does not require an account for evaporation as the latter plays a minor role in the hydrological balance which might result in a landslide (Brand et al., 1984; Chatterjea, 1989; Morgenstern, 1992).

Thick silty covers may instead experience a hydrological behaviour strongly affected by evaporation. In principle, modelling evaporation requires coupling water and heat flows. Geotechnical engineers and geologists are still too unfamiliar with heat flow modelling in particular, as it entails a number of thermal parameters and boundary conditions that are difficult to calibrate and validate. In addition, governing equations need non-widespread numerical codes and their high non-linearity involves difficulties in achieving numerical solutions. This implies that in several applications evaporation is neglected at all as it was considered less important than rainfall intensity during a highly intense event that triggered landslides. (e.g., Baum et al., 2008; Pagano et al., 2010; Formetta et al., 2016) or taken in to account following approaches with different degree of complexity (Casadei et al., 2003; Rosso et al., 2006; Šimunek et al., 2006; Ebel et al., 2010; Formetta et al., 2014; Capparelli and Versace, 2011; Arnone et al., 2011). Complete approaches, modelling internal and boundary evaporation through hydrothermal approaches, were taken into account in studies referred to slopes in fine-grained soils differing substantially from those involved in the case in hand (Cui et al., 2005; An et al., 2017; Song et al., 2016). Concerning the hydrological behaviour of silty volcanic sloping covers, several Authors adopted approaches incorporating evaporation for the interpretation of monitoring results (Pirone et al., 2015a) and/or back-analysis of previous events (Greco et al., 2013; Napolitano et al., 2016). In such studies, however, evaporative fluxes were modelled as a boundary phenomenon only.

The question naturally arises whether, for silty volcanic sloping covers, the accuracy of the early warning prediction will be significantly reduced if evaporation is neglected, resulting in too many false alarms.

This study attempts to address this question by comparing results yielded by three different mathematical-numerical models, either taking evaporation into account as only superficial or also internal phenomenon (Wilson et al., 1994) or neglecting it (Richards, 1931), in the interpretation of the 2005NIL case study.

Two of the models account for evaporation: one based on a coupled (heat-water flow) approach, and the other based on an isothermal approach. The third model neglects evaporation entirely. It represents an update to the approach previously adopted in Pagano et al. (2010). Suction and other hydrological variables predicted by using all the selected models are presented and discussed in an attempt to characterize their various performances.

These three cited models are presumed to be operating in real time, namely receiving recorded and/or forecasted meteorological variables as input data and returning variables relating to slope safety conditions as output data. To this aim, they need to be applied to geomorphological contexts that may be modelled assuming in 1D flow conditions, to save as much analysis time as possible (Pagano et al., 2010; Greco et al., 2013). The paper also discusses which simplifications are able to accelerate predictions without excessively reducing their accuracy.

Considerable effort has been made in this study to develop original procedures for calibrating model parameters from the interpretation of the experimental results provided by the above-mentioned lysimeter. Model parameters are typically quantified in laboratory at a scale much smaller than field conditions. In this perspective, a strength point of the paper is represented by the use, during the calibration and validation procedure, of the findings retrieved by the above-mentioned lysimeter, involving 1 m$^3$ of material forced by realistic boundary conditions provided by actual meteorological evolution, instead of the traditional procedures based on small specimen subject to artificial boundary conditions. A previous comparison among laboratory, lysimeter and field conditions, restricted to retention properties of the same soil involved in the work (Pirone et al., 2016) resulted in a satisfactory agreement. This encourages throughout the work considering representative lysimeter of field conditions also for quantifying parameters needed to describe soil hydraulic conductivity and thermal behaviour.

The paper begins with a description of the case study and presents the lysimeter data. After describing the selected models and simplifications carried out to save analysis time, it illustrates all the procedures followed to calibrate the parameters. Lastly, it presents and compares the results of the analyses, discussing model performance from the point of view of their possible use as early warning predictors.

## 2 Methods

### 2.1 Field experimental data: the Nocera Inferiore 2005 landslide

2005NIL involved a triangular shaped area of 24,600 m$^2$ and a soil mass of 33,000 m$^3$ covering a 36° open slope (Fig. 1c). In the uppermost part of the landslide, in the triggering zone, the slope angle approaches 39° (de Riso et al., 2007) and the pyroclastic cover is made up of 2 m thick loose non-plastic silty sand (volcanic ash) (Fig. 2). The bedrock consists of highly fractured limestone located at a depth ranging from 1 to 2 m, approaching the maximum values at the apical zone.

The landslide triggered in the apical zone, spreading downward. The rapid post-failure movement caused the death of three people whose house was destroyed by the impact of the soil mass, which then covered a wide area (20,000 m$^2$) at the toe of the slope (Fig. 1c). In the same zone, two smaller landslides occurred less than 1 km from the main one at the same time.

For this case-study, in addition to hourly precipitation values, the availability of air relative humidity and air temperature records makes it possible to estimate the evaporative fluxes potentially experienced by the cover involved in the landslide, complementing precipitation in characterizing the fluxes that have affected the hydrological state of the cover over time. Data are retrieved by a weather station located very close to the investigated slope (Figure 1a). They have an hourly resolution and refer to the timespan 1 January 1998-1 November 2008.

Figure 3 plots the evolution of precipitation, air temperature and air relative humidity, at the landslide site over a time span of about ten years (1998-2008), including the investigated landslide occurred on 4 March 2005 (precipitation until the landslide time are reported in Pagano et al., 2010). Changes in daily air temperature (Fig. 3a) and air relative humidity (Fig. 3b) are used to estimate the daily potential evaporation (Fig. 3d) by following the FAO guidelines (Allen et al., 1998). The potential evaporation intensities occur much lower than precipitation intensities (Fig. 3c). However, potential evaporation persistency makes cumulated values (Fig. 3e) significant even during winter, when the evaporative flux is minimum.

Figure 3e shows that the hydrological year in which the landslide took place is associated with the highest cumulated precipitation. The most significant spread between cumulated values of precipitation (1200 mm) and potential evaporation (380 mm) is observed at the time of the landslide (see vertical segment with rows in Fig. 3e).

For the site here considered, monitoring of field hydrological variables were not available so to allow a characterization of safety conditions based on field measurement.

## 2.2 Experimental data provided by the physical model

An extensive description of the physical model and the experimental data used can be found in Rianna et al. (2014a, b). A wooden tank (Fig. 4) houses a 0.75 m thick layer of non-plastic sandy-silt volcanic soil. This soil was selected and placed in such a way as to try to reproduce the intrinsic properties and field porosities (around 70%) of the material involved in 2005 NIL, which grain size distribution is reported in Fig. 2. A geotextile bounds the bottom of the layer. As its voids are larger than those of the overlying soil, it acts as a capillary barrier, namely, an impervious boundary as far as suction remains higher than zero, and a draining surface just when suction vanishes (Reder et al. 2017). The behaviour at the bottom should therefore be consistent with that of the fractured bedrock with partly empty fractures or a gravel layer (pumice).

The monitoring system implemented in the physical model (Fig. 4) makes it possible to obtain potential fluxes (total precipitation and potential evaporation), actual fluxes developing across the uppermost layer surface (actual evaporation and infiltrated precipitation), and the effects induced by these fluxes within the layer (suction, volumetric water content, temperature). Fluxes are quantified by a meteorological station, the continuous weighting of the layer (by three load cells sustaining the tank) and monitoring of all the energetic terms involved in the energy exchanges between the soil and atmosphere

(by radiometer, pyrometer, heat flux plate, thermistors). Matric suctions (by jet-fill tensiometers and heat dissipation probes), volumetric water contents (using TDRs), and soil temperatures (using thermistors) are monitored at four depths within the layer.

The physical model was exposed to the atmosphere over four hydrological years in bare conditions, and it returned a number of behavioural patterns for the evolution of the layer's hydrological and thermal states. These patterns suggest the ingredients that a predictive model should include, allow quantification of soil properties, and they ultimately represent a useful reference framework to assess reliability of numerical predictions.

The reference behavioural pattern considered here is represented by the evolution of water storage (WS) in the layer (Fig. 5), soil suction (at two depths – Fig. 6a), and temperature (three depths – Fig. 7) over four hydrological years (the first two years of WS and suction records are by Rianna et al., 2014a; the first two years of temperature records are by Rianna et al., 2014b). WS, expressed in terms of overall water volume in the layer divided by the layer surface (Fig. 5a), increases at the onset of wet periods due to precipitation (Fig. 5b) infiltrating the layer. For prolonged wet periods, such as those occurring during the first, third, and fourth years, WS tends to stabilize at a wet level (Fig. 5a, "wet threshold" line) placed just below the maximum saturation value in the layer (Fig. 5a, "saturation" line). Above this wet level, drainage is often observed during, and immediately after, rainfall events, whereas drainage has never been observed below it. During dry periods, water storage decreases due to evaporation (Fig. 5c), and during prolonged dry periods it tends to reach a minimum at the dry threshold. Soil suction (Fig. 6a) measured at different depths is consistent with water storage evolution (Rianna et al., 2014a). It progressively increases during the dry period, resulting in asynchronous fluctuations, indicating a slow propagation through the sample of the changes in the atmospheric conditions. Suction values exceed the jet-fill tensiometer full scale during the summer periods, but reduce to few kPa during the wet season, resulting in this case in synchronous fluctuations, in line with prompt propagation throughout the layer of changes in boundary conditions determined by the atmosphere.

Temperature (Fig. 7) measured at the four depths follows an evolution consistent with atmospheric temperature. Fluctuations in the temperature of the atmosphere tend to reduce as depth increases due to greater soil filter action.

## 2.3 Predictive models

Three different models (Fig. 8) were selected to convert the meteorological evolution (Fig. 3) recorded at the landslide area into hydrological variables for the cover. Shared features are:

- the one-dimensionality of water fluxes; this hypothesis, formulated in order to save analysis time in early warning applications, should also lead to realistic estimations of the hydrological state of the cover in line with indications of previous works; in particular, according to Pagano et al. (2010) this hypothesis is reliable for homogeneous cover, as 1D and 2D numerical analyses yield in this case very comparable results; reliability should be extended also for the frequent cases of homogeneous layers resting on pumices, as pumices effects may be reduced to suitable boundary conditions (Reder et al., 2017); Papa et al. (2009) observed the one-dimensionality of the water flux also throughout layered volcanic slopes, by plotting water flux vectors obtained from field monitoring of suction at the site of Monteforte Irpino; also Damiano et al. (2017)

confirmed that one-dimensionality of water fluxes is common throughout layered volcanic slopes; on similar geomorphological contexts, Pirone (2009) and Napolitano et al. (2016) model the actual stratigraphy retrievable on field; the former brings to light that extreme wet and dry periods result into 1D (respectively downward and upward) flow conditions throughout the cover and that transition periods associate with rotation of flow vectors;

- the rigidity of the domain; neglecting the effects of deformational processes induced by suction changes is suggested by the stiff volumetric response observed under swelling-reloading paths, which should speed up re-equilibrium processes so that hydraulic response is unaffected by the consolidation delay;

- modelling of hourly rainfall as hydraulic boundary conditions at the top surface of the domain; there a water flux condition equal to rainfall intensity is maintained if pore water pressure at the top is less than zero (a positive value would indicate ponding formation); otherwise, a null pore water pressure condition is assumed and maintained if the computed entering water flux is less than rainfall intensity; on the contrary, entering flux is again applied at the rainfall intensity.

- surface seepage is applied at the bottom boundary; this condition corresponds to the hypothesis that the bedrock the volcanic layer rests on is intensely fractured and that the fractures are filled only by air (Reder et al., 2017).

The more comprehensive approach adopted couples the water balance equation with the heat balance equation and thermodynamic equilibrium (Wilson et al., 1994) and is applied using the Vadose/W code (GeoSlope, 2008).

The water balance equation is expressed as:

$$\frac{1}{\rho_w g} \frac{\partial (u_a - u_w)}{\partial t} = \frac{1}{\rho_w g m_2^w} \left[ \frac{\partial}{\partial z} \left( k_w + \frac{k_w}{\rho_w g} \frac{\partial (u_a - u_w)}{\partial z} \right) + \left( \frac{P_a + u_v}{P_a \rho_w} \right) \frac{\partial}{\partial z} \left( D_v \frac{\partial u_v}{\partial z} \right) \right] \tag{1}$$

where $u_w$ (M L$^{-1}$ T$^{-2}$) = liquid pore water pressure, $u_a$ (M L$^{-1}$ T$^{-2}$) = pore air pressure, $u_v$ (M L$^{-1}$ T$^{-2}$) = partial pressure of vapor pore water, $m_2^w$ (L T$^2$ M$^{-1}$) = slope of soil water characteristic curve (SWCC), $P_a$ (M L$^{-1}$ T$^{-2}$) = total atmospheric pressure, $k_w$ (L T$^{-1}$) = hydraulic conductivity function (HCF), $D_v$ (T) = function of vapor diffusivity through the soil, $\rho_w$ (M L$^{-3}$) = liquid water density, $g$ (L T$^{-2}$) = gravitational acceleration.

In comparison with the traditional form of water balance equation, describing the flow of liquid water through porous media, equation (1) contains an additional term (the second one in square brackets) considering possible changes in the water phase.

The heat balance equation is expressed as:

$$C_h \frac{\partial T}{\partial t} = \frac{\partial}{\partial z} \left( \lambda \frac{\partial T}{\partial z} \right) - L_v \left( \frac{P_a + u_v}{P_a} \right) \frac{\partial}{\partial z} \left( D_v \frac{\partial u_v}{\partial z} \right) \tag{2}$$

where $T$ (Θ) = soil temperature, $C_h$ (ML$^{-1}$T$^{-2}$Θ$^{-1}$) = function of volumetric specific heat, $\lambda$ (MLT$^{-3}$Θ$^{-1}$) = function of thermal conductivity, $L_v$ (L$^2$T$^{-2}$) = latent heat of water vaporization.

In this equation, the last term accounts for the amount of energy spent on water vaporization and represents the coupling with the water balance equation.

Thermodynamic equilibrium is expressed as:

$$u_v = u_{v0} \, exp\left(\frac{(u_a - u_w)M_w g}{RT}\right)$$

(3)

where $u_{v0}$ (M L$^{-1}$ T$^{-2}$) = saturated partial pressure of pore vapour, $M_w$ (M N$^{-1}$) = water molecular weight, $R$ (ML$^2$T$^{-2}$N$^{-1}$Θ$^{-1}$) = ideal gas constant.

The described model requires the following boundary conditions:

- soil suction $(u_a - u_w)_s$ or, alternatively, liquid water flux $(v_w)_s$ at the top-boundary; infiltrating precipitation was reproduced as already described in this section, and actual evaporation $AE$ was reproduced according to the FAO approach described in what follows; $u_a$ is assumed to be the same as the atmospheric pressure;

-vapor pressure $(u_v)_s$ or, alternatively, vapor water flux $(v_v)_s$; the boundary value problem was addressed by quantifying the former from air relative humidity $RH$ and air temperature $T_a$ records; $RH$ provides the ratio $(u_v)_s/(u_{v0})_s$, while $T_a$ provides the partial pressure of the vapor phase in saturated conditions $(u_{v0})_s$ using the Tetens equation (Tetens, 1930);

- temperature $T_s$ is assumed to equal air temperature $T_a$ measured two meters above the surface of the ground, in line with the approach followed by Wilson et al. (1997).

Maximum values of Actual evaporation $AE$ correspond to atmospheric demand Potential evaporation $PE$; however, according soil moisture conditions AE can be reduced according to the falling law proposed by Wilson et al. (1997)

$$AE = k \, PE$$

(4)

where k is equal to:

$$k = \frac{exp\left(\frac{(u_a - u_w)_s M_w g}{RT_s}\right) - RH}{1 - RH}$$

(5)

and the $PE$ expression of the FAO approach (Allen et al., 1998):

$$PE = k_{crop}\left[\frac{0.408 \, \Gamma \, (R_n - G) + \eta \frac{900}{T_a + 273} u_{2m}\left(u_{v0}^{air} - u_v^{air}\right)}{\Gamma + \eta(1 + 0.34 u_{2m})}\right]$$

(6)

where $k_{crop}$ (-) = crop coefficient, $\Gamma$ (ML$^{-1}$T$^{-2}\Theta^{-1}$) = slope of the vapor pressure curve, $\eta$ (ML$^{-1}$T$^{-2}\Theta^{-1}$) = psychrometric constant, $R_n$ (ML$^{-2}$T$^{-3}$) = net radiation flux, $G$ (ML$^{-2}$T$^{-3}$) = soil heat flux, $u_{2m}$ (L T$^{-1}$) = wind speed measured two meters above the surface of the ground. The FAO approach takes the various crop conditions into account thanks to the $k_{crop}$ coefficient that transforms the potential evaporation of the reference surface (in square brackets) into $PE$ in relation to the actual surface (a bare surface in the case at hand). This coefficient was quantified in Rianna et al. (2014b) as $k_{crop} = 1.15$ by using $PE$ measurements provided by the physical model. This is consistent with the literature indications (Allen et al., 1998; Allen et al., 2005).

It is important to highlight that the model described above incorporates evaporation as both a superficial and an internal phenomenon, suited to reproducing the possible deepening of the water state-change surface over dry and hot periods. Hereafter, this model will be referred to as the "Internal Evaporation Model" or "IEM." Such modelling has been successfully adopted for investigating different issues: e.g. embankment stability analysis (Gitirana, 2005; Briggs et al., 2016), soil-structure interaction (Al Qadad et al., 2012), generic soil-water budget (Cui et al. 2005).

Richards equation constitutes a simplified version of the described model. It consists of the water balance equation (1) for liquid water, thus removing the vapor flow term, and equation (4), specified by equation (5) and (6). This isothermal approach only takes into account changes in atmospheric temperature through eq. (5), where it is assumed that $T_s = T_a$ (an approach called "isothermal model with atmospheric coupling", by Fredlund et al., 2012). The model takes only evaporation into account as a boundary phenomenon, which occurs only at the top-boundary, with no possibility of a downward shift of the water state-change surface. Hereafter, this model will be referred to as the "Boundary Evaporation Model" or "BEM".

A further simplification often adopted in single applications (Pagano et al., 2010) or in a number of codes developed to estimate slope stability conditions over extensive sloping territory (Iverson, 2000; Montgomery and Dietrich, 1994; Baum et al., 1998) corresponds to the latter approach (Richards equations) applied without accounting for evaporation. This model will subsequently be referred to as the "No Evaporation Model" or "NEM".

All the simulations were performed using an input/output hourly time step; moreover, the adaptive time stepping scheme proposed by Milly (1982) was adopted for inner time step.

## 2.4 Calibration and model parameters

The IEM model is a generalization of the BEM or NEM models. Hence, the soil properties and parameters IEM contains also pertain to BEM and NEM. Calibration refers to both thermal (soil thermal conductivity and volumetric specific heat) and hydraulic properties (the soil water retention curve and the hydraulic conductivity function), the latter being common to the other two approaches.

Parameters were calibrated from the results provided by the physical model over the first two years. The results collected over the subsequent two years were adopted to validate the calibration.

Some of the soil properties were quantified directly from measurements. This is the case of the SWCC, which was obtained as best fit function of (water content) – (suction) points recorded by TDRs and jetfill tensiometers (Fig. 9a). The SWCC adopted interpolates experimental data available up to suction values of around 90 kPa and extrapolates them to much higher suction levels, adopting as asymptote the lowest volumetric water content measured locally by TDRs.

5   This is also the case of the thermal conductivity function ($\lambda$), which was derived as best-fit function (Fig. 9b) of (water content) – (thermal conductivity) points recorded at all depths. Thermal conductivity values were obtained using heat dissipation probes, referring to the relationship relating probe energization $q$, to $q$-induced temperature changes $\Delta T$ (Shiozawa and Campbell, 1990):

$$q = 4\pi \ln(\Delta T)\,\lambda \qquad\qquad (7)$$

which contains $\lambda$ as a single unknown.

The volumetric specific heat $C_h$ and hydraulic conductivity functions, on the other hand, were quantified following articulated and novel interpretation procedures of experimental data based on back-analysis.

$C_h$ was determined by solving the heat equation at sublayers with the thicknesses demarcated by a triplet of temperature
measurement points. By assuming that the energy spent for evaporation within each sublayer is negligible, and $\lambda$ is constant at the mean value assumed throughout the sublayer ($\bar{\lambda}$), the heat equation may be rewritten as:

$$\frac{\partial T}{\partial t} = \frac{\bar{\lambda}}{C_h}\frac{\partial^2 T}{\partial z^2} \qquad\qquad (8)$$

This equation was solved by regarding the two external temperature measurements of the sublayer as boundary conditions and
considering internal measurement $T^*$, a reference for calibrating $\bar{\lambda}/C_h$, as the value that gives the best fit of the evolution of $T^*$ over the first two years of observations. Preliminary knowledge of $\bar{\lambda}$ then makes it possible to obtain $C_h$ (Fig. 9c).

Having established SWCC and thermal properties, the HCF was calibrated via a back-analysis of the hydrological behaviour of the layer observed over the first two years. A domain of the same thickness (0.75m) as the layer placed in the physical model was assumed to be subject to boundary conditions at the top surface reproducing the recorded atmospheric variables.
Figure 10 shows that the agreement between measurements and predictions obtained from the back-analysis is satisfactory both in terms of water storage (Fig. 10a) and suction (Fig. 10b). Predictions match observations not only within the calibration range but also outside it (validation phase). Figure 11 plots the hydraulic conductivity function resulting from back analysis. The hydraulic conductivity of the material drops by three orders of magnitude (from $10^{-6}$ to $10^{-9}$ m/s), with suction increasing from 0 to 100 kPa.
Table 1 lists parameter values obtained from calibration based on experimental results provided by the physical model. Hydraulic conductivity function and water retention curve are fully consistent with those indicated for the same soil type by

Nicotera et al. (2010) and Pirone et al. (2015b). Thermal parameters for these soils are instead quantified in this work for the first time.

Upon completion of parameter calibration, thermal functions $\lambda$ and $C_h$ were further validated by checking their ability to reproduce the observed thermal behavior under the effects of the recorded atmospheric variables. Figure 12 shows that the model yields temperature predictions fully consistent with temperature measurements at all depths over the four years of available observations.

As stated previously, IEM calibration also implies automatic BEM and NEM calibration. Figures 13a and 13b add to the plots of Figure 10 the predictions of the BEM model in terms of WS and suction. It can be observed that even the approach incorporating evaporation as only a boundary phenomenon performs well in reproducing the recorded hydrological pattern. The predictions yielded by the two approaches (IEM and BEM) match during winter and early spring, in the meanwhile that actual evaporation attains potential levels, and evaporation really configures as a boundary phenomenon. They depart instead during late spring, summer and autumn, when internal evaporation phenomena take place, or the effects of their previous occurrence are felt. During these periods, IEM predictions reproduce observations better than BEM ones, as the latter either overestimates water storage or underestimates suction.

A synthetic overview about the capability of calibrated approach in reproducing observed values is reported in Table 2. It reports an evaluation of goodness-of-fit for WS, suction and temperature obtained by using Nash-Sutcliffe (Nash and Sutcliffe, 1970), Kling-Gupta (Gupta et al., 2009) and the coefficient of determination. Advantages and constraints of such approaches are largely investigated in literature (Krause et al., 2005; Bennet et al., 2013). The results are differentiated considering calibration and validation periods both separately and jointly. In general, satisfying results are achieved for WS and temperature where over all periods the three indices never fall below 0.74 (over 0.88 for WS); on the other hand, worse performances arise considering suction, especially adopting Nash-Sutcliffe approach. These discrepancies are likely to be due to disregarding hydraulic hysteretic dynamics using a unique SWCC aimed to catch the average behaviour of soil in wetting and drying paths. As shown by Rianna et al., 2017, this simplification may sometimes result in an increased time-lag of suction-drop response to precipitation, especially when the initial state point falls near the main drying curve.

## 3 Results and discussion

### 3.1 Patterns of AE predictions under simplified PE evolutions

In order to further investigate differences in the predictions yielded by the IEM and BEM models, the model responses were compared under much simpler boundary conditions than those considered in reproducing the behaviour of the physical model (Fig. 13). With the models calibrated as described above, the two approaches incorporating evaporation were compared in a numerical experiment. They were used to predict actual evaporative fluxes (AE) they return under the effects of the same virtual potential evaporation (PE) flux applied at the top boundary. PE is supposed to act with an intensity of 4.5 mm per day and constant over 60 days. Such value represents the mean evaporative atmospheric demand estimated through available

weather forcing for summer during the physical model monitoring. The response of the two models was plotted in terms of both AE evolution (Fig. 14a) and hydraulic conductivity evolution at the top surface (Fig. 14b). The AE values yielded by the two models coincide when AE is equal to PE, in line with what was illustrated above in discussing the trends in Figure 13. There is a time, under such forcing, when AE departs from PE in the two predictions. It corresponds to the moment when

upward water fluxes are no longer able to fully supply the top surface with the water amount satisfying the hypothesized atmospheric demand for evaporation (4.5 mm per day). After this time, the two AE predictions also diverge and follow a substantially different evolution. A drop in AE typifies the BEM model response. This is due to the fact that the model generates vapour only from liquid water reaching the top surface. When AE diverges from PE, a shallow thin zone placed below the top surface desaturates more than the interior. With increasing suction, hydraulic conductivity decreases and hydraulic gradients

increase, with the former having a greater effect on flow reduction (Fig. 14b). In turn, this reduction slows the upward water flux, so that that water losses induced by the PE action in this shallow thin zone are no longer compensated by water supplied from the interior. In this way, under the effect of desaturation dynamics the shallow-thin zone shortly reaches the residual water content and very low hydraulic conductivity values. At this point, the upward water flux from the interior is almost totally inhibited, in a sort of barrier effect exerted by the shallow thin zone, implying the vanishing of AE. This barrier effect

inhibits further desaturation processes inside the domain. Consistently, suction attains very high values within the cap zone, remaining at low levels inside the domain.

On the other hand, the IEM model is suited to reproduce the occurrence of internal evaporation as soon as the zone placed below the top surface desaturates more than the interior at the time of PE-AE divergence. Vapour starts to generate within the interior of the domain other than at the boundary, in an automatic and progressive deepening of the water phase-change surface.

Interior vapour forms and migrates upward at a rate that is now regulated by vapour conductivity. This latter, unlike with hydraulic conductivity, progressively increases as the degree of saturation decreases, so that vapour migration takes place at a rate that maintains AE at significant levels. It follows that AE reduction is gradual in the IEM predictions as it is effectively supplied by the interior vapour generation. Furthermore, as internal vapour approaches the top-surface, it also reduces suction levels there as a consequence of thermodynamic equilibrium (Eq. 3). This contributes to maintaining hydraulic conductivity

at levels much higher than those associated with the BEM prediction. Boundary evaporation also occurs at a consistently higher rate, or, in other words, the formation of any barrier effect is prevented. Compared with BEM prediction, in the IEM prediction, vapour migration produces suction levels (Fig. 15c) that are lower near the top surface and higher within the domain interior when desaturation processes are more significant.

It is worth highlighting that the high suction values predicted by the above numerical analyses at the top-surface are not only

30 theoretically based (see for example Wilson et al., 1994; 1997), but they could be experienced by silty pyroclastic covers during the dry-hot season. For instance, Pagano et al. (2014) measured suction at the layer topsoil by heat dissipation probes up to 10000 kPa. Both numerical and experimental results hence indicate that during the hot-dry season very high topsoil suction merge with moderately lower interior suction values.

### 3.2 Interpretative analyses of the Nocera Inferiore 2005 landslide

#### 3.2.1 Preliminary assumptions and considerations

In order to investigate the potential of the IEM and BEM models in interpreting the 2005NIL and establishing their reliability as predictive tools in early warning systems, the cover thickness (2 meters) in the zone where the landslide triggered was analysed, by assuming one-dimensional water fluxes.

The reliability of the one-dimensional assumption has already been discussed at the beginning of §2.3. The analyses were carried out according to the hypotheses formulated in §2.3, quantifying the model parameters from experimental data provided by the lysimeter (Table 1). As mentioned in the Introduction section, hydraulic retention properties estimated from laboratory and lysimeter experiments were compared with those obtained by field monitoring (Pirone et al., 2016). Consistency of results encouraged in using lysimeter-based quantification of all parameters for field predictions. The "specimen" adopted in the present study to quantify parameters ($1m^3$) is five orders of magnitude larger than those typically adopted ($3\ 10^{-5}m^3$). Hydraulic conductivity is hence measured at a mesoscale, a condition that should be quite representative of field conditions.

The chain of events inducing a rainfall-induced landslide in a silty volcanic sloping cover can schematically be illustrated in subsequent or parallel stages as follows (Pagano et al., 2008): (1) a generalized suction drop, (2) a suction-induced strength reduction, (3) trigger of instability due to local peculiarities, internal or external to the slope, (4) propagation of local trigger throughout the cover. The approach followed in interpreting the 2005NIL in view of structuring an early warning prediction consists in limiting the analysis to the characterization of suction levels predisposing to instability (stage (1)), rather than carry out all steps towards an assessment of the slope safety factor. Converting suction distribution into prediction of slope safety factor would involve a slope stability analysis. This latter implies the complex issue of characterizing soil strength and the strength action exerted by roots, which is very difficult to quantify. In addition, the uncertainty in a local trigger cause would intrinsically make difficult carrying out stability analyses.

As root strength action also founds on suction levels (root-soil bonds are mostly promoted by suction), the approach followed limits to predict the suction-dependent predisposing stage by physically based approaches. WS may also be used in place of suction, as it is a function of suction distribution. It has already been adopted as proxy of slope safety conditions in previous early warning applications, starting from the well-known Seattle implementation (Baum and Godt, 2010).

The evolution of meteorological variables recorded at the landslide site over the whole ten years (Fig. 3), including the time of the landslide, was then converted into the evolution of WS and suction in the middle of the domain analysed (to a depth of 1 m) via the IEM and BEM predictions. To the aim of possibly using these models for physically based early predictions, these variables were considered as proxies from which different alert thresholds can be defined.

#### 3.2.2 Results

Figure 15a plots the WS evolutions yielded by the IEM and BEM analyses. Both are able to reproduce the typical WS patterns that develop over a hydrological year, so as shown by the experimental data provided by the physical model: WS fluctuates

during the year, increasing during autumn, winter and early spring due to precipitation, and reducing during late spring and summer due to evaporation.

The IEM and BEM trends differ systematically during the dry seasons as a result of the different AE mechanisms that activate as soon as significant drying processes cause AE to diverge from PE (see §3.1). The minimum WS values predicted by the two models during the dry periods are different, as already observed in the results meant to fit the data for the physical model (Fig. 13a). However, differences in minimum values are now higher than those previously computed, essentially due to the different domain thicknesses analysed (2 m instead 0.75 m). The 0.75 m domain is so small that the meteorological forcing typical of the dry periods produces water content to attain residual value throughout the whole thickness. Most of the drying processes (WS reductions) are regulated by AE=PE. Only a residual part of the WS (from the point of departure down to the minimum) occurs because of internal evaporation. Under these conditions, IEM and BEM effectively work in a similar way most of the time, producing a small gap at the lowest WS. A 2-meter thick domain, under the effects of similar meteorological forcing, does not achieve residual water content at all depths during the dry season. WS losses regulated by internal evaporation are far from minor and represent a significant part of the evaporated water, so that a significant gap is present at the lowest WS. This is of around 150 mm for the years characterized by matching WS during the wet season.

WS gaps attained during the dry season are reflected in WS differences during the subsequent wet season (autumn), when landslide susceptibility is usually moderate. These differences tend to be gradually attenuated with the passing of time due to the higher potential infiltration in the IEM domain caused precisely by its drier state: the WS trend yielded by IEM is marked by higher hydraulic gradients. In some seasons, IEM predictions always remain below those of the BEM, while in others the WS gaps disappear. The occurrence of one or the other condition depends essentially on rainfall cumulating over the autumn wet season. When the IEM prediction and the BEM match, AE=PE usually occurs, so that IEM and BEM predictions coincide over the entire subsequent wet period. This happened during the four consecutive hydrological years from September 2002 to the August 2006 landslide, including the landslide year.

A peak at record is yielded by both IEM and BEM analyses at the time of the landslide (Fig. 15a). By enlarging the scale and looking at high WS values in order to isolate and highlight peaks (Fig. 15b), it may be observed that the peaks attained at the time of the landslide are higher than the peaks attained over other years. This encourages the assumption that both predictions, as they clearly indicate a peculiarity in the hydrological response in the cover at the time of the landslide, would have worked satisfactorily if they had been adopted as predictive tools in an early warning system. In fact, they would have been able to indicate a situation of alarm at the time of the landslide without generating a significant number of false alarms if the alarm threshold had been placed slightly below the peak attained at the landslide time.

Figures 15c,d show the IEM and BEM predictions for the evolution of suction at a depth of 1 m. They reveal that suction predicted also at the middle of the layer may work as a proxy for slope safety conditions, as the hydrological behaviour it depicts is consistent with everything indicated by the integral variable WS. Suction or WS may therefore both be used as reference variables for early warning.

Figures 15a,b show the water storage evolution yielded by the simplest model (NEM) adopted, which neglects evaporation entirely (Pagano et al., 2010; Reder et al., 2017). In this case, it is not possible to carry out a continuous analysis of the hydrological response of the cover over the whole ten years, due to the inability of the model to predict water losses from the domain during the dry periods. Analysis needs to be restarted at the beginning of each hydrological year in order to reinitialize

the hydraulic variables. Unfortunately, these become additional input data that have to be set. It would be necessary, in theory, to monitor suction or water content in the field to quantify initial conditions. The initial quantification of suction has a particular impact on the reliability of the analysis during the periods of landslide susceptibility occurring not far from the start time (considered at the beginning of the hydrological year). The effects of the starting conditions are in fact lost after a period of around four months (Pagano et al., 2010). In these predictions, suction value at the beginning of each hydrological year should

be provided by field monitoring. Due to unavailability of monitoring data, suction re-initialization at the beginning of each hydrological year is achieved by adopting suction values yielded by the IEM model. The NEM prediction has a peak at the time of the landslide, once again at record levels both in terms of water storage and suction. The peak is however not as high as a large number of other significant peaks. From the point of view of performance, NEM by itself must therefore be considered less effective than the IEM and BEM predictions together.

The performances of the three adopted approaches can be judged by taking the number of alerts and alarms they would have yielded if they had been adopted as predictive tools in an early warning system within the specific reference period analysed. Differences in performance obviously depend on the levels at which the thresholds for water storage or suction are set to limit the different alert stages and spread the alarm. The alarm threshold can be identified by interpreting the landslide phenomenon by the models, referring to the prediction yielded by the most complete one (IEM) in terms of the lowest suction level not

associated with a landslide (around 3.5kPa). A possible pre-alarm alert level is 5 kPa, obtained by increasing the alarm threshold by around 50%. This double choice permits a quantitative comparison of model performance, with the IEM model performing the best, returning 0 alarms and 1 alert (1 every 11 years). The BEM comes second, returning 3 false alarms (1 every 3.7 years) and 5 alerts (1 every 2.2 years). The NEM model performs worst, returning 5 false alarms (1 every 2.2 years) and 12 alerts (0.9 for year).

**4 Conclusions**

This paper has investigated the performance of three physically based models taken from the literature with a view to using them for early warning predictions. Two of them incorporate evaporative fluxes, but the other neglects them. Particular care has been taken with the simplification of the models so as to respect the accuracy of the predictions they provide, establishing procedures to calibrate the parameters and to characterize the hydrological patterns they predict. The models' performance has

been assessed by using them to interpret the case history of a landslide and examine their ability to indicate any hydrological peculiarity at the time of the landslide.

In analysing model performances, this work structures an entire procedure for early warning prediction. It bases on the following key points:

(1) the assumption of WS or, alternatively, suction as proxies of slope safety evolution, overcoming the huge problems of characterizing complex strength factors and dealing with aleatory and undetectable local conditions that might generate landslide trigger;

(2) the quantification of soil parameters at a mesoscale level, by interpreting experimental results from a 1 m$^3$ large layer subject to realistic boundary conditions, rather than, as usual, from small laboratory specimens subject to artificial boundary conditions; in particular, the paper suggests an experiment typology and elaborates theoretical interpretation procedures for parameter calibration; for all contexts similar to that here analysed (homogeneous layers made of soils similar to that investigated), parameter values provided are ready for use and hence, they could greatly simplify the task of setting an early warning prediction;

(3) the key assumption of one-dimensionality of water flux, able to save time by preserving, at the same time, prediction reliability; in general, the comparison between typical thickness of quite homogeneous pyroclastic covers and slope length make reliable the 1D assumption, even if the features of actual flow might locally depart from those ideally hypothesised, due to local inhomogeneity difficult to detect or specific hydrological conditions of lateral diversion; these local conditions, although they could represent a cause for local triggering, should however not affect the average suction levels that, yielded by a 1D hypothesis, predispose the slope to propagate a local triggering.

The study shows that all models taken into consideration, if used as early warning predictive tools, would then be able to signal the alarm at the time of a landslide. Increased complexity and completeness of the models, however, would clearly result in a lower number of false alarm predictions.

The calibrated models, along with the detected thresholds, may be adopted as predictive tools in early warning systems for the numerous slopes with similar physical characteristics (slope gradient, slope thickness, intrinsic and state soil properties, vegetation), requiring site monitoring of, at least, hourly precipitations, daily air temperature and relative humidity.

The following limitations should however be highlighted, in order to prevent from referring the same procedure to inappropriate contexts:

(a) at present, the procedure may be confidentially applied to homogeneous sloping layers made of non-plastic silty volcanic soils; the possibility for extending it to other contexts has to be preliminarily investigated; premises encourage for extending the procedure to all contexts susceptible to rainfall-induced landslides

(b) the one-dimensional assumption is validated for a homogeneous layer; this approach is quite established in literature but in layered contexts the proposed procedure should be carefully validated;

(c) the hydraulic hysteresis of the soil has been neglected in the present study, by assuming a unique SWCC fitting all the available observations over the calibration time span; the accuracy loss of the prediction due to this simplification is at present topic of new research;

(d) the assumed lowermost boundary condition (seepage surface) might be not realistic in contexts differing from that here considered. For instance, fractured formations filled with fine material transported from downward flux should be more appropriately analysed by assuming a unit gradient or infinite layer condition.

As concerns the setting of early warning thresholds, the implementation of the procedure could face, in several cases, with the unavailability of meteorological evolutions associated with landslides. In such a case, first trial thresholds in terms of suction or WS might be fixed conservatively on the base of the lowest suction levels (highest WS) ever predicted in converting the real meteorological evolution. On the other hand, among the several meteorological evolutions available some of them could relate with a landslide. It is worth noting at this point that even a single well-documented case, as it was for the present study, represents a very lucky circumstance. The first trial alarm thresholds may then be based on suction or WS levels predicted at landslide time. In both circumstances (availability or not of a landslide case-study), initial calibrated thresholds should be continuously validated and, eventually, updated over time as the site of interest could be subject to substantial geomorphological modifications affecting, in increase or decrease, its susceptibility to rainfall induced landslides.

For monitored sites, a more accurate calibration/validation procedure could also be based on fitting monitored quantities.

**Acknowledgements**

Alfredo Reder and Luca Pagano have partly developed this work within the framework of the PRIN 2015 project titled "Innovative Monitoring and Design Strategies for Sustainable Landslide Risk Mitigation". They wish to thank the Ministero dell'Istruzione dell'Universitá e della Ricerca Scientifica (MIUR) which have funded this project.

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

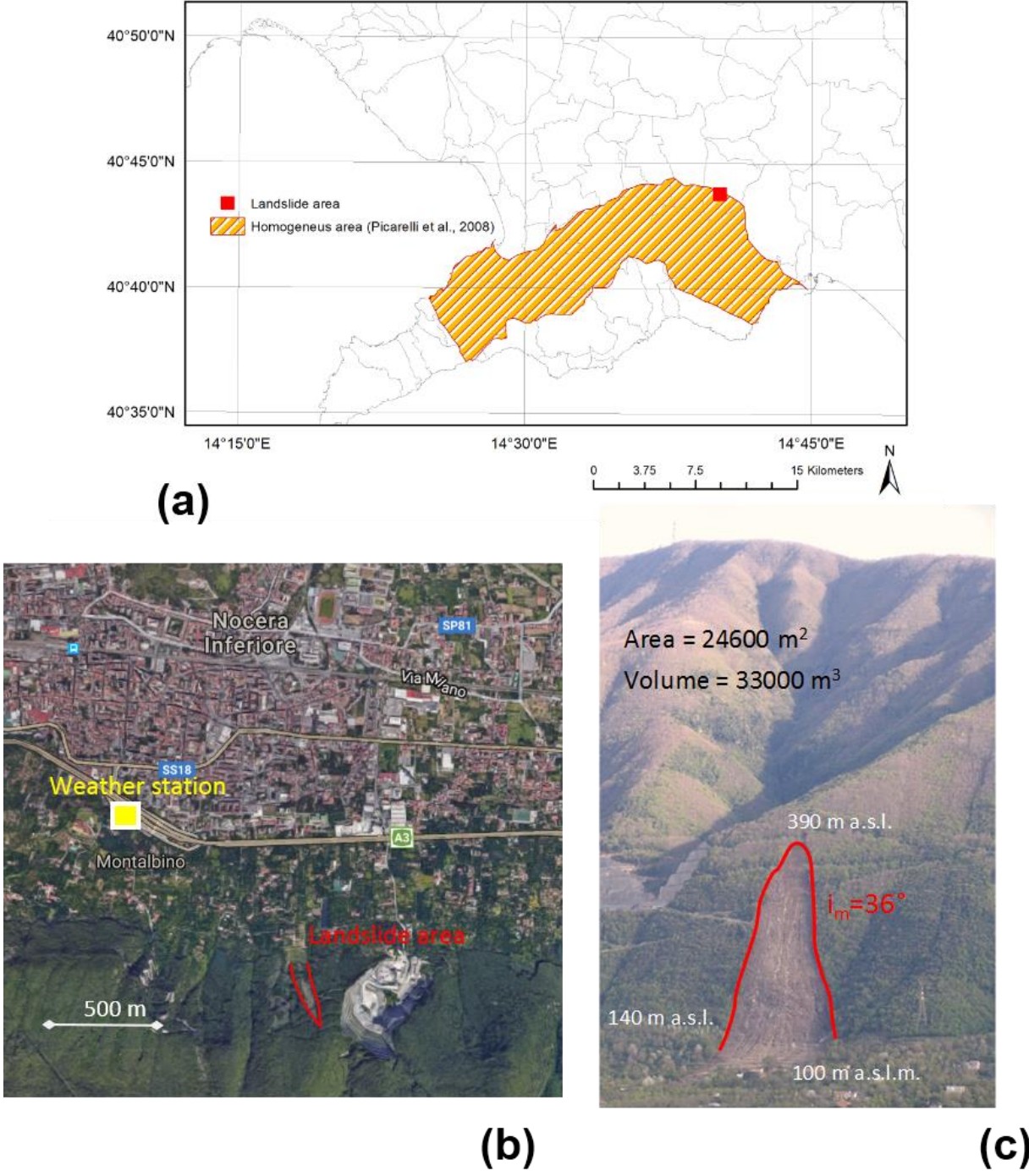

**Figure 1: The 2005 Nocera Inferiore Landslide (2005NIL): (a) map indicating the landslide location and zones homogeneous with that of the landslide for soils, cover thicknesses and slope gradients; (b) plan view (DigitalGlobe 2012. http://www.earth.google.com), indicating the landslide area and the location of the weather station; (c) frontal view of the landslide area. (Pagano et al., 2010, modified).**

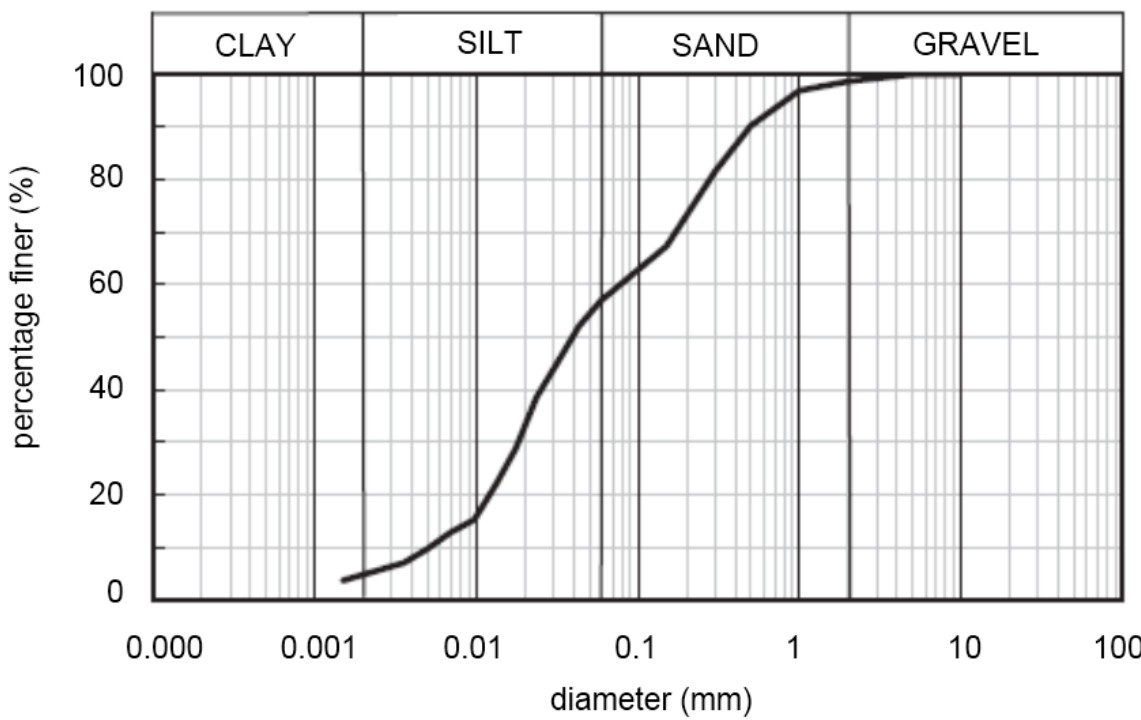

**Figure 2: Grain-size distribution of the Nocera Inferiore volcanic ash (Pagano et al., 2010)**

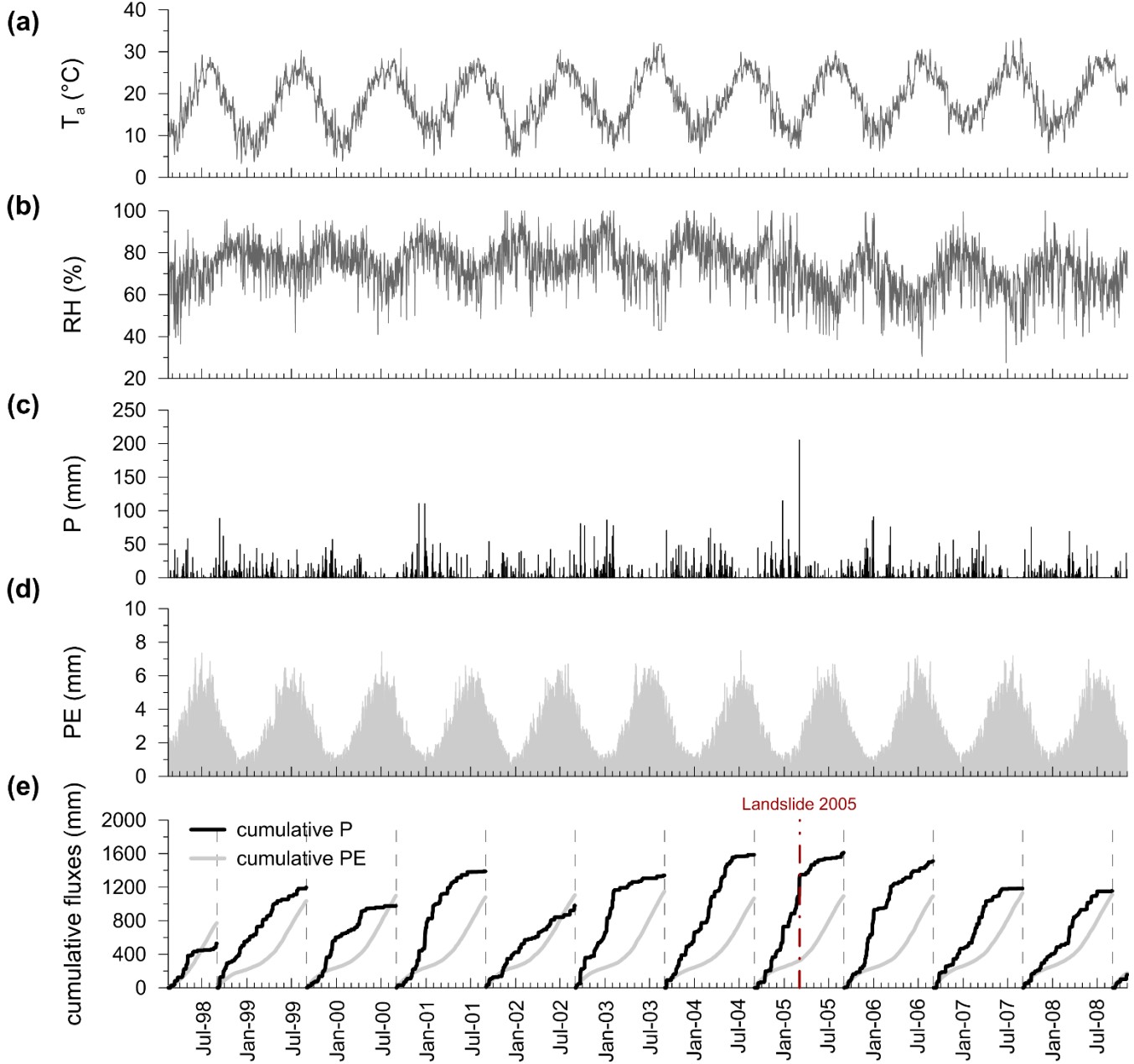

**Figure 3: Meteorological variables recorded at the 2005NIL site between January 1998 and August 2008: (a) mean daily air temperature, $T_a$; (b) mean daily air relative humidity, RH; (c) daily precipitations, P; (d) daily potential evaporation, PE; precipitation, P, and potential evaporation PE, cumulated during each hydrological year**

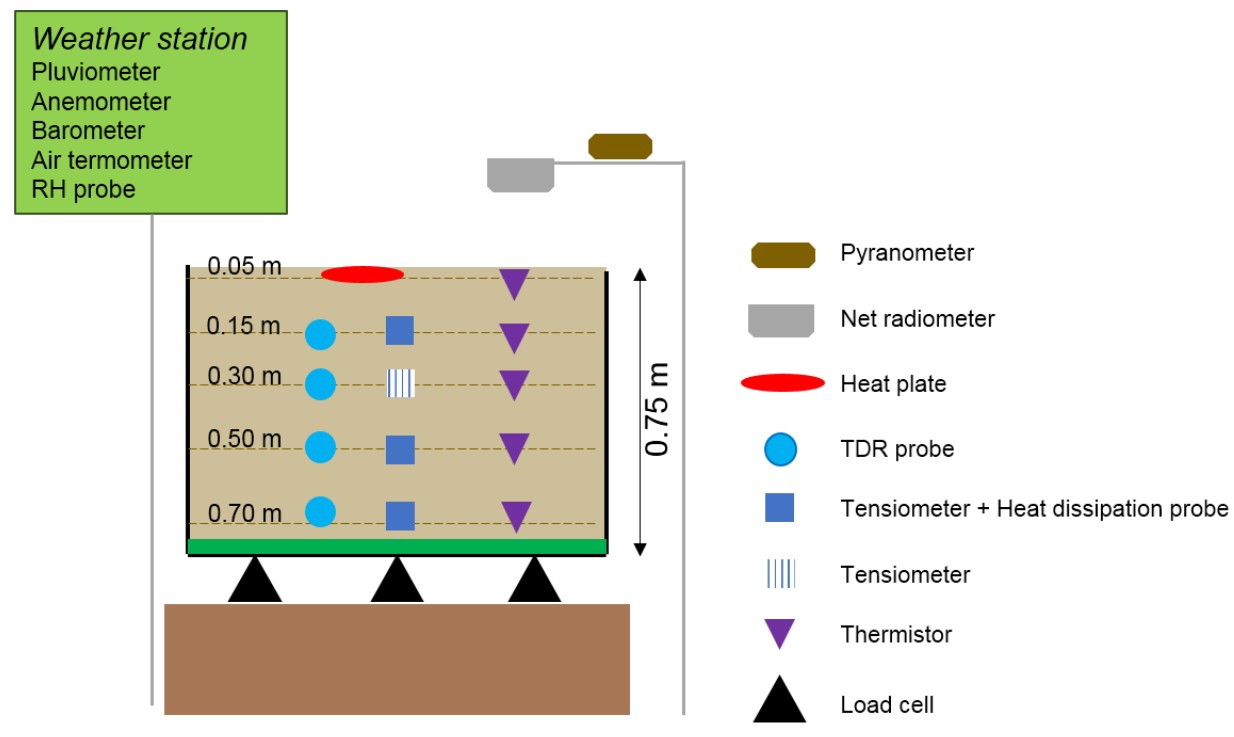

**Figure 4: Monitoring devices installed in the physical model (Rianna et al., 2014a, modified).**

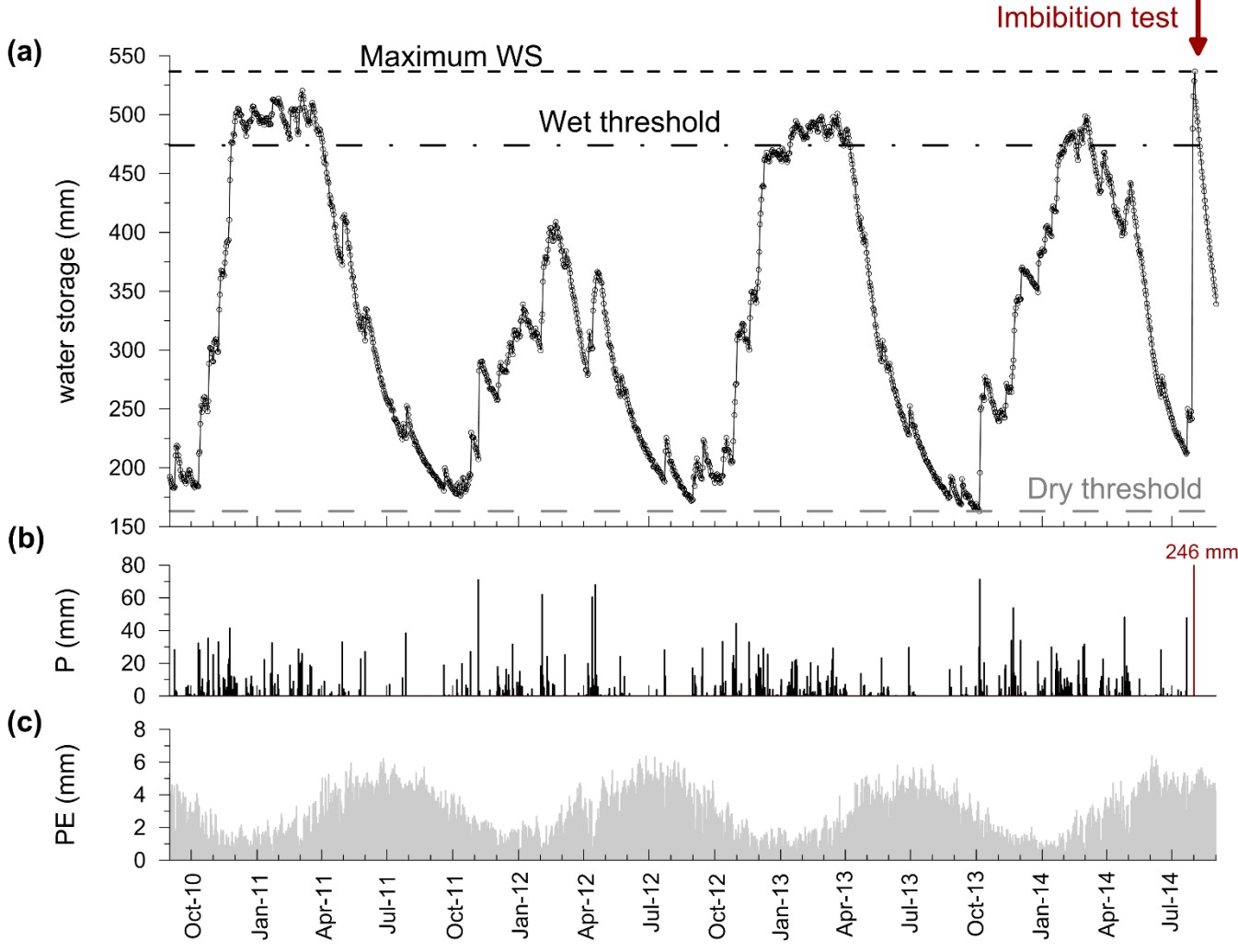

**Figure 5: Evolution of variables recorded by the physical model of lysimeter: (a) layer water storage, WS; (b) daily precipitations, P; (c) daily potential evaporation, PE. (Records taken over the first two hydrological years are from Rianna et al., 2014a).**

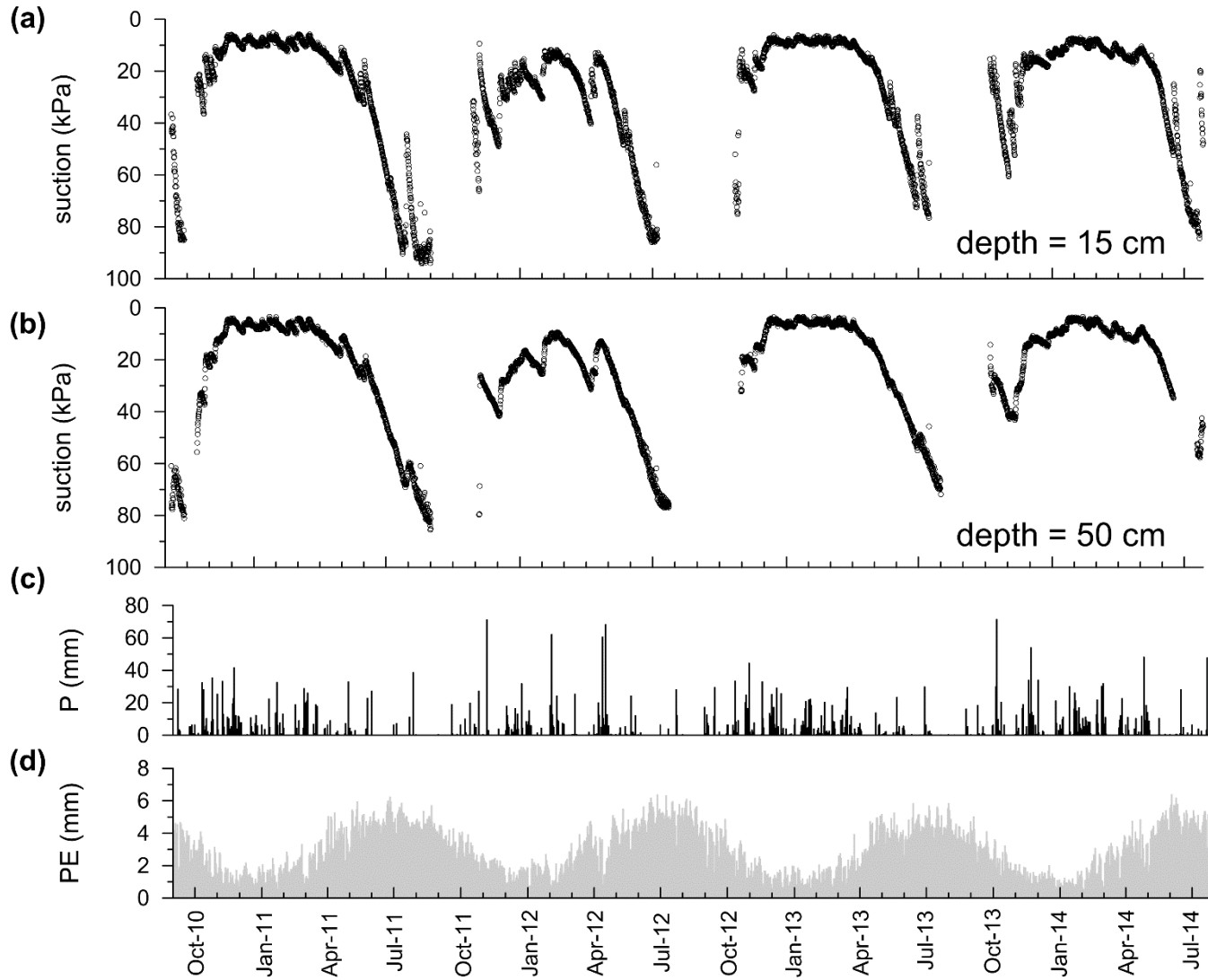

**Figure 6: Evolution of variables recorded by the physical model of lysimeter: (a) matric suction at depth=15 cm; (b) matric suction at depth=50 cm; (c) daily precipitations, P; (d) daily potential evaporation, PE. (Over the first two hydrological years records are from Rianna et al., 2014a)**

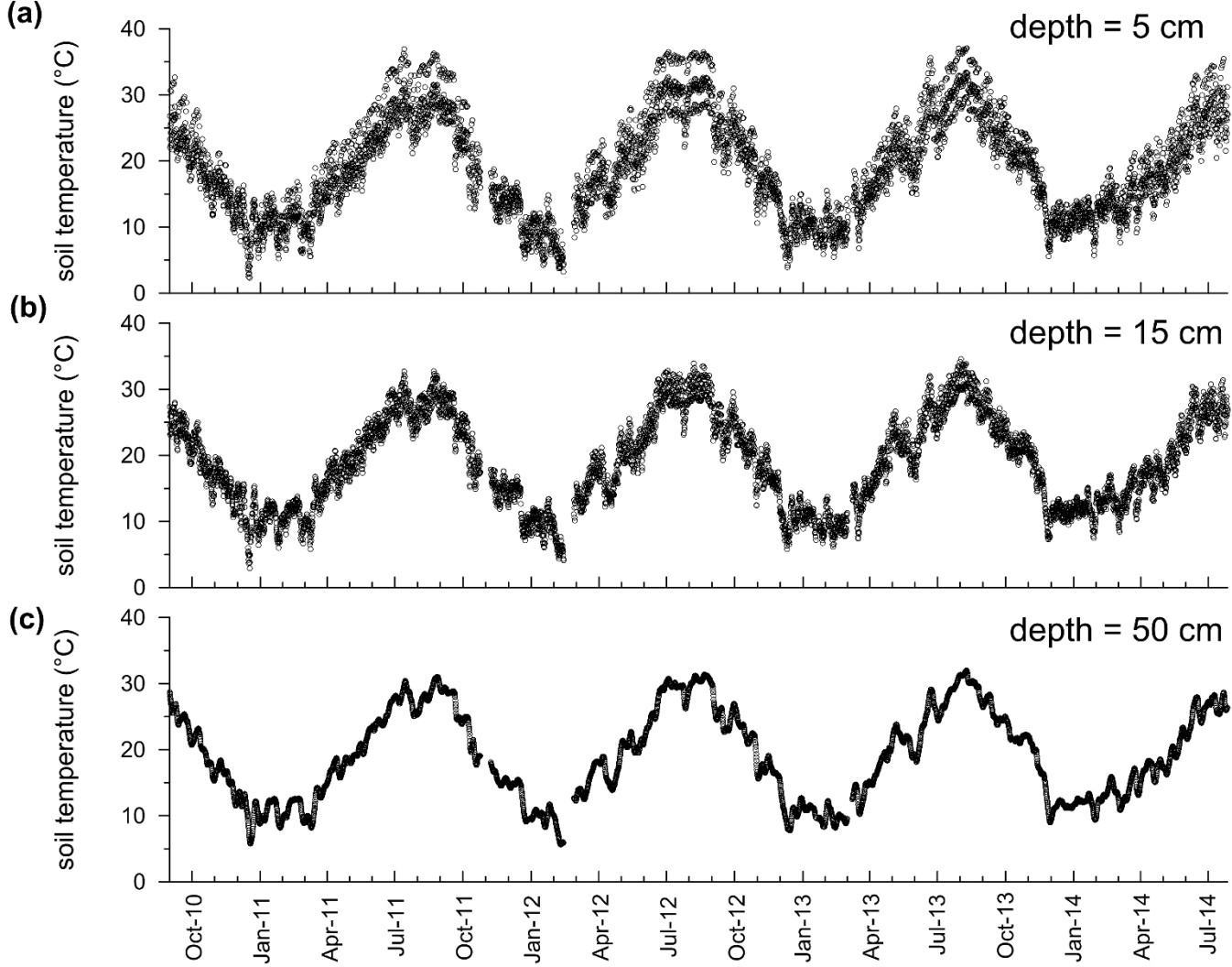

**Figure 7: Evolution of temperature, T, recorded by the physical model of lysimeter: (a) soil temperature at depth=5 cm; (b) soil temperature at depth=15 cm; (c) soil temperature at depth=50 cm.**

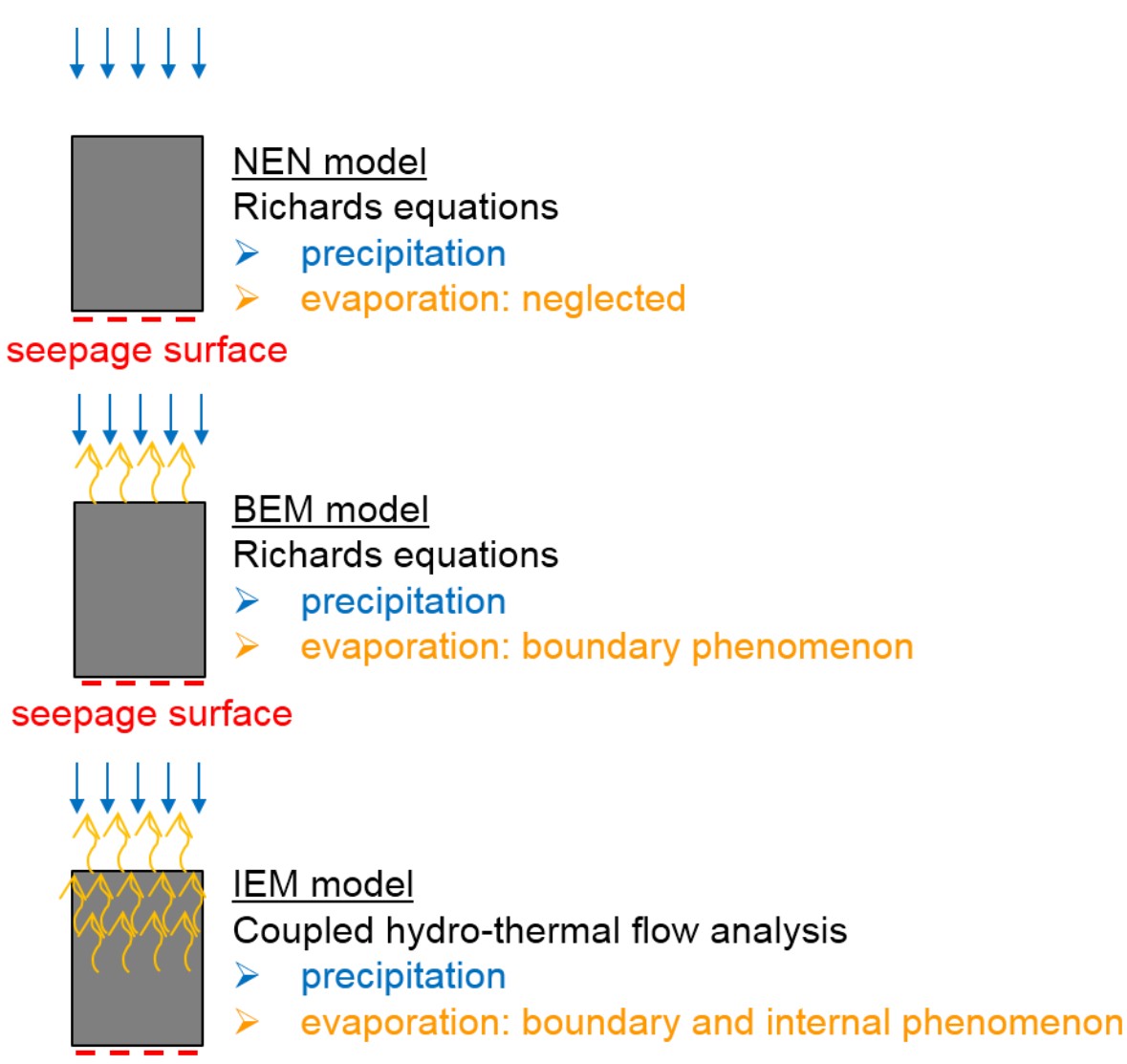

**Figure 8: Model schemes adopted for early warning predictions.**

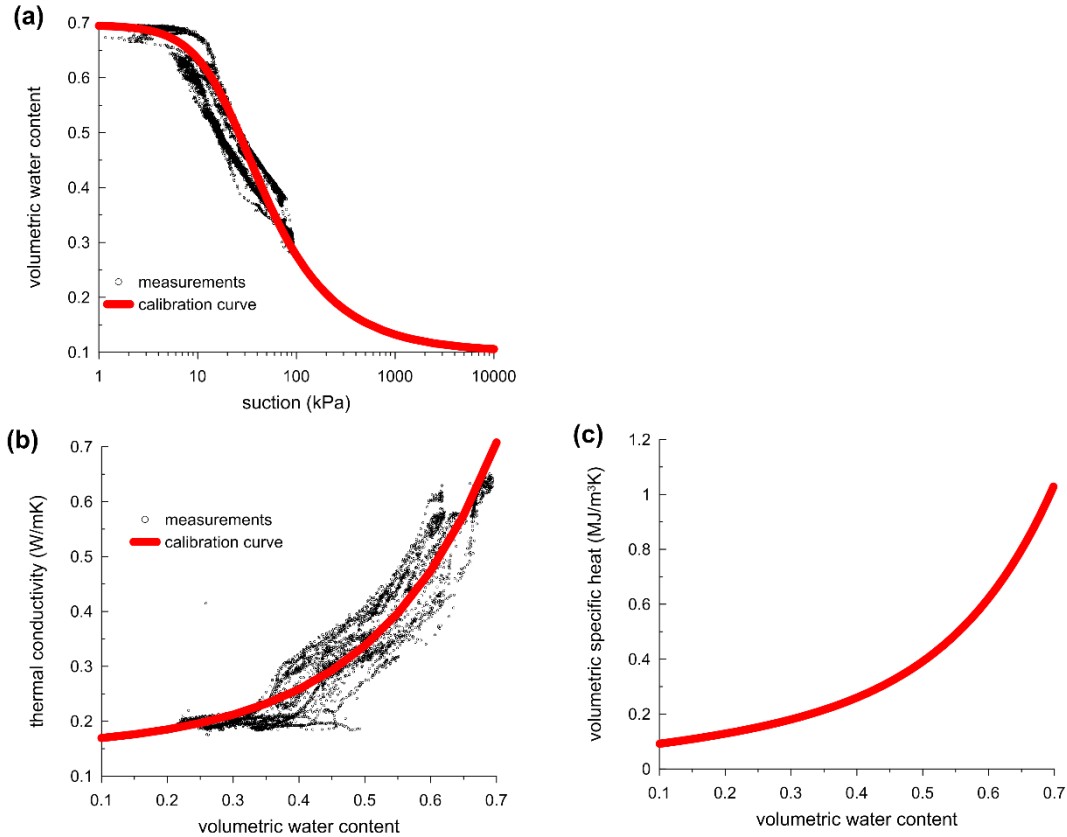

**Figure 9: Calibration of model parameters: (a) experimental point at the depths of 15 and 70 cm  versus fitting water retention curve; (b) experimental point versus fitting thermal conductivity function; (c) volumetric specific heat function**

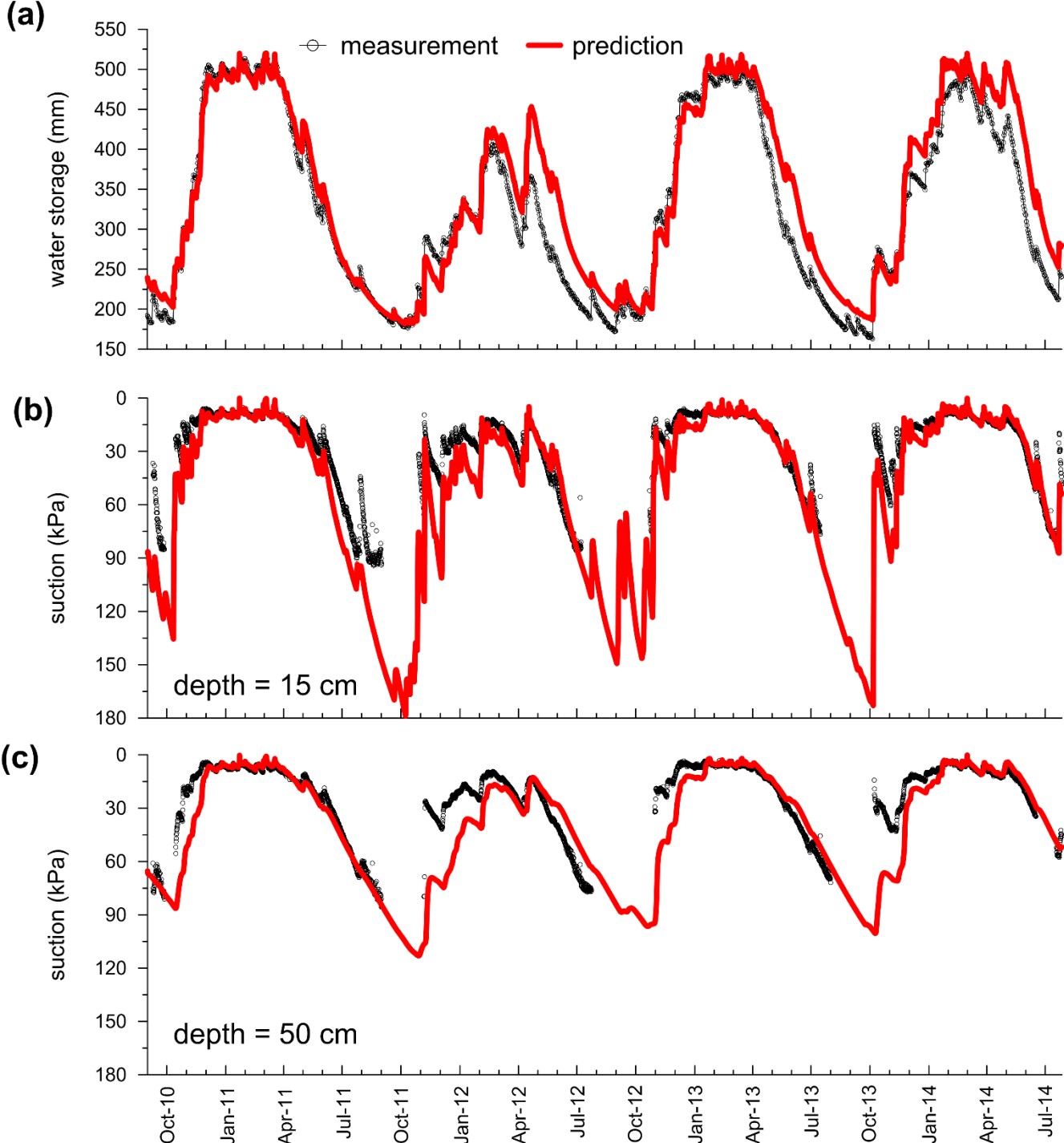

**Figure 10: Back-analysis of the observed layer hydrological behavior by the IEM model: (a) measured versus IEM-predicted water storage, WS; (b) measured versus IEM-predicted suction at depth=15cm; (c) measured versus IEM-predicted suction at depth=50cm.**

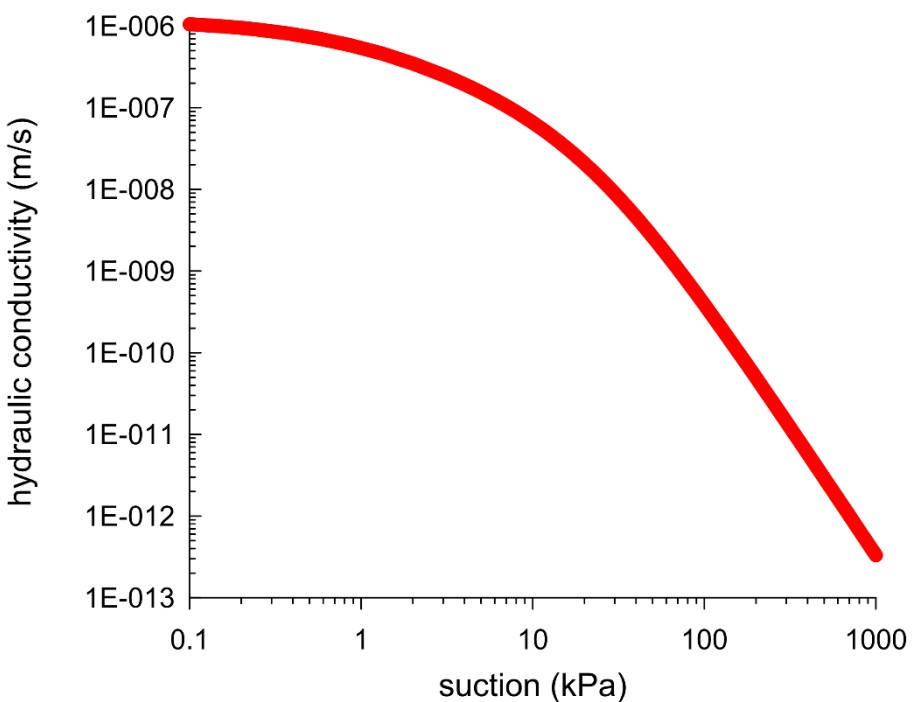

**Figure 11: The hydraulic conductivity function obtained from the IEM-back-analysis of the hydrological behaviour.**

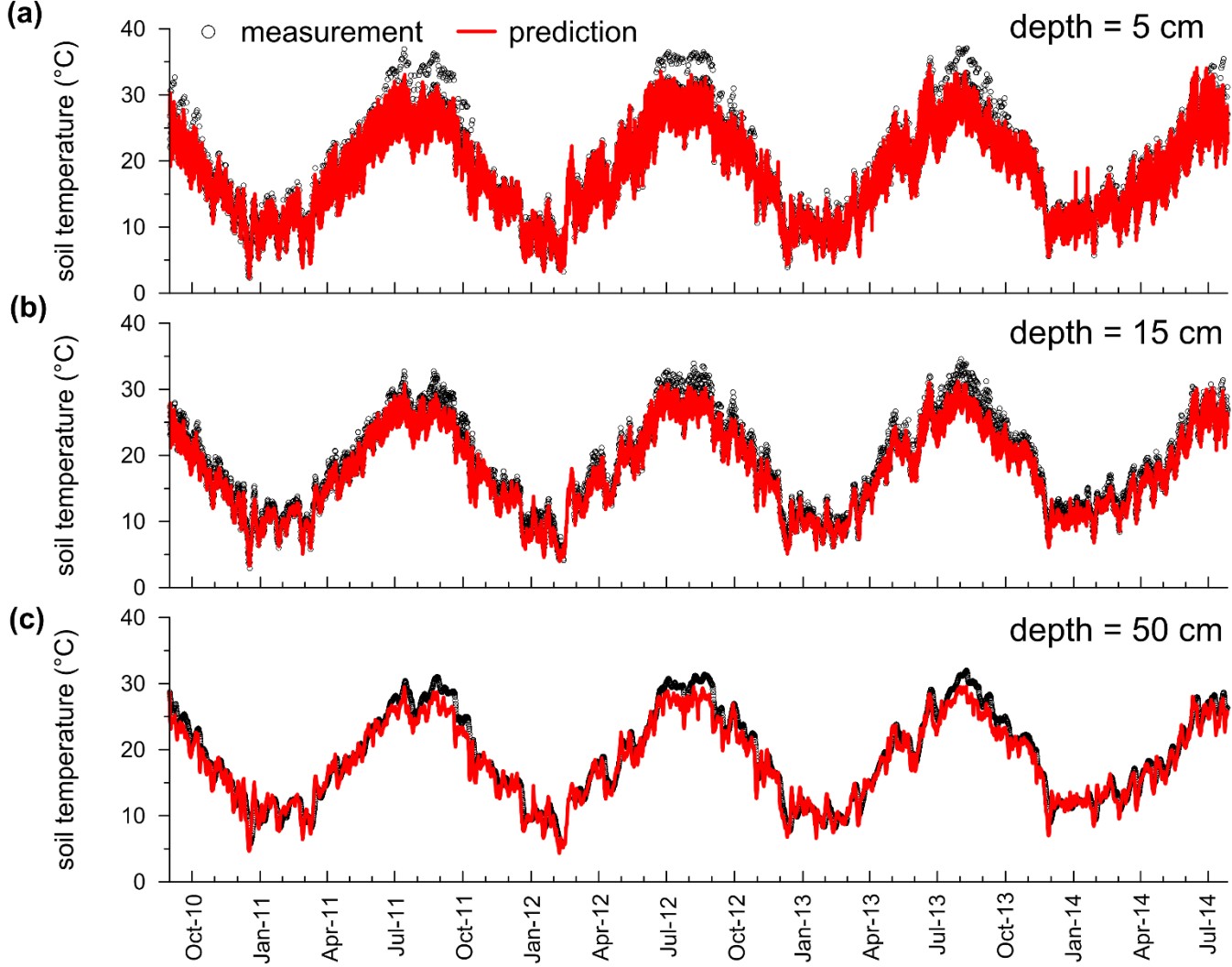

**Figure 12: Measured versus IEM-predicted temperature (T) evolutions at three different depths indicated.**

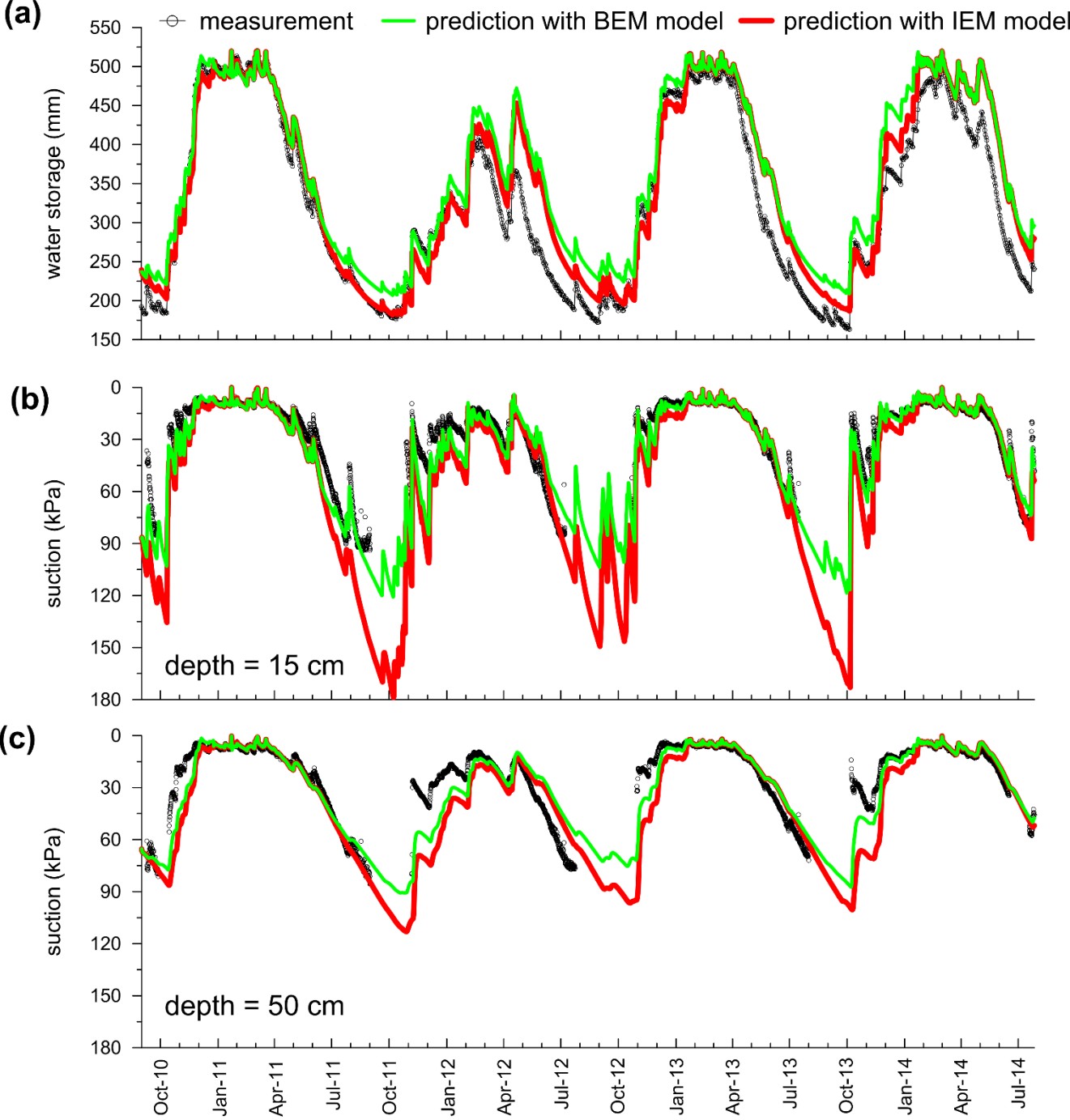

**Figure 13: Comparison between IEM-predicted, BEM-predicted and observed hydrological behaviour: (a) comparisons in terms of water storage, WS, evolutions; (b) comparisons in terms of suction evolutions at depth=15cm; (b) comparisons in terms of suction evolutions at depth=50cm.**

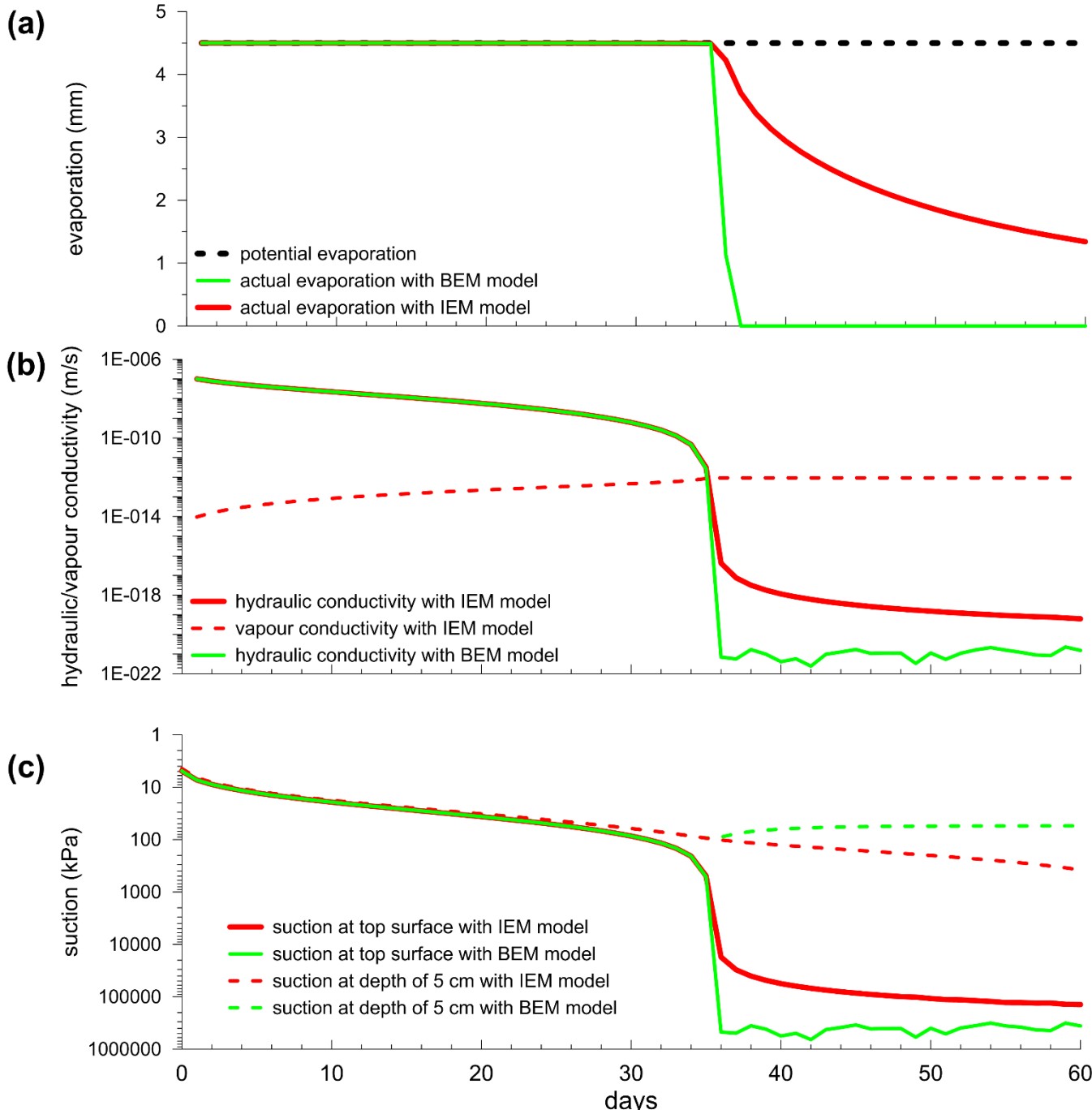

**Figure 14: Comparison of variables yielded by IEM and BEM predictions under PE=4.5 mm/day: (a) comparisons between actual evaporative fluxes, AE; (b) comparisons between hydraulic conductivities at the top-surface and evolution of vapor conductivity at the top-surface; (c) comparisons between evolutions of suction at the top surface and at depth=5cm.**

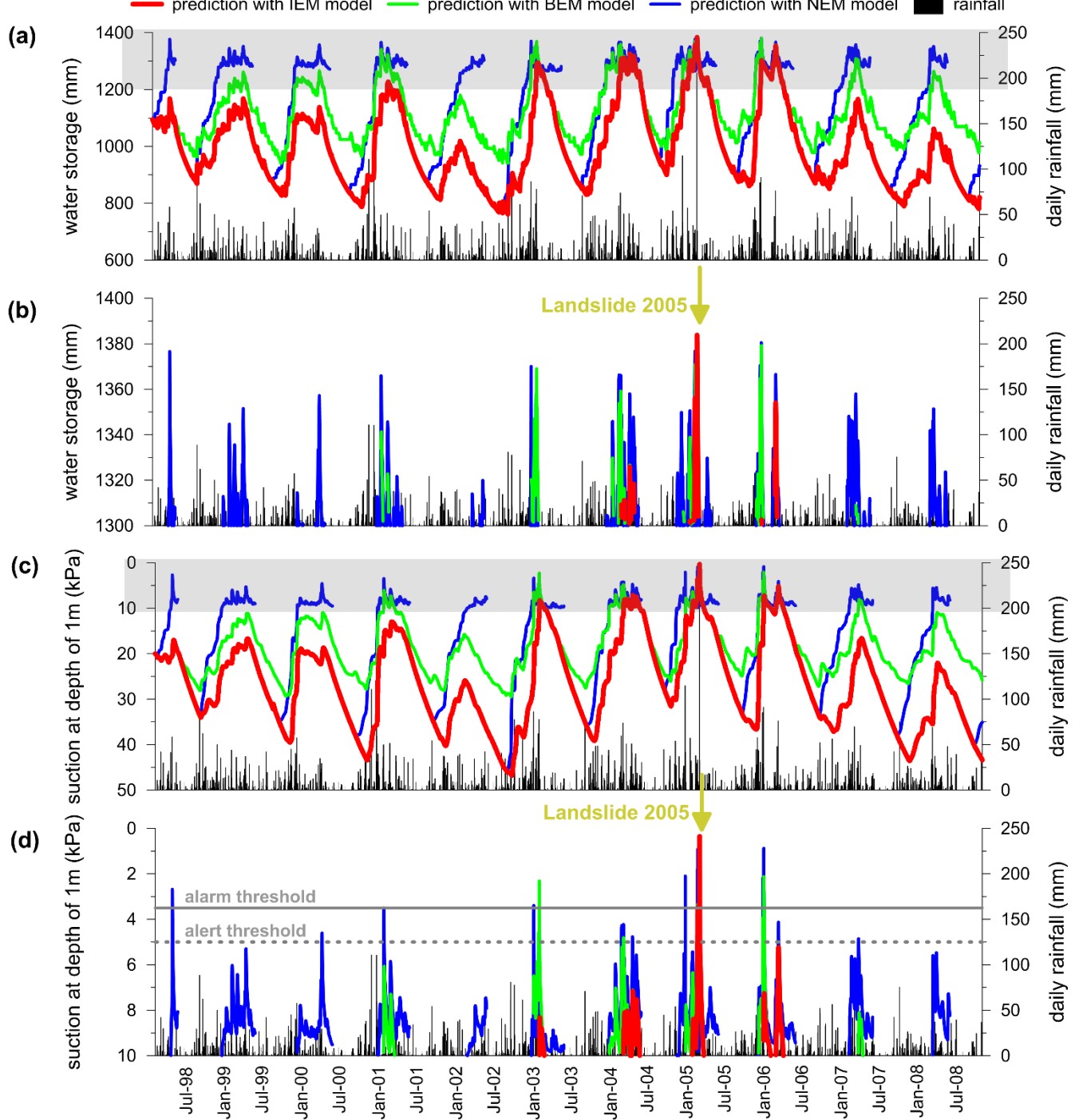

**Figure 15: NEM, BEM and IEM predictions of the hydrological behaviour evolution for the layer involved in the 2005NIL: (a) prediction within the WS full range; (b) prediction for WS levels higher than 1300 mm; (c) prediction within the suction full range; (d) prediction for suction levels lower than 10 kPa.**

**Table 1. Soil functions and estimated parameters for investigated soil**

| Soil water characteristic curve | Hydraulic conductivity function | Volumetric specific heat function | Thermal conductivity function |
|---|---|---|---|
| $\dfrac{\theta - \theta_r}{\theta_s - \theta_r} = S_e = [1 + (\alpha\,\lvert u_a - u_w \rvert)^n]^{-\left(1 - \frac{1}{n}\right)}$ *(Van Genuchten, 1980)* $S_e$ = effective saturation degree $\theta_s$ = residual water content = 0.696 $\theta_r$ = saturated water content = 0.100 $\alpha$ = inverse of air entry suction = 0.047 1/kPa $n$ = fitting parameter = 1.760 | $K_w = K_{ws}(S_e)^l \left\{ 1 - \left[ 1 - (S_e)^{\frac{1}{m}} \right]^m \right\}^2$ *(Mualem, 1976 - Van Genuchten, 1980)* $K_{ws}$ = saturated hydraulic conductivity = $1 \times 10^{-6}$ m/s $l$ = fitting parameter = -0.5 | $C_h = C_{h0} + \alpha_0 exp(b_0\theta)$ $C_{h0}, \alpha_0, b_0$ fitting parameters $C_{h0}$ = 0.055 MJ/m$^3$K $\alpha_0$ = 0.025 MJ/m$^3$K $b_0$ = 5.252 | $\lambda = \lambda_1 + \alpha_1 exp(b_1\theta)$ $\lambda_1, \alpha_1, b_1$ fitting parameters $\lambda_1$ = 0.148 W/mK $\alpha_1$ = 0.013 W/mK $b_1$ = 5.406 |

**Table 2. Nash-Sutcliffe, Kling Gupta and Coefficient of determination assessed for proxy variables (WS, suction and temperature at different depths) considering all period, only calibration period (First and Second Hydrological Year), only validation period (Third and Fourth Hydrological Year)**

|  | Nash-Sutcliffe | | | Kling Gupta | | | Coefficient of determination | | |
|---|---|---|---|---|---|---|---|---|---|
|  | All | Cal | Val | All | Cal | Val | All | Cal | Val |
| WS | 0.89 | 0.91 | 0.88 | 0.93 | 0.94 | 0.92 | 0.93 | 0.93 | 0.93 |
| Suction 15 cm | 0.54 | 0.43 | 0.68 | 0.58 | 0.53 | 0.72 | 0.85 | 0.86 | 0.83 |
| Suction 30 cm | 0.57 | 0.55 | 0.55 | 0.72 | 0.71 | 0.73 | 0.74 | 0.75 | 0.69 |
| Suction 50 cm | 0.36 | 0.45 | 0.04 | 0.65 | 0.71 | 0.50 | 0.61 | 0.62 | 0.54 |
| Suction 70 cm | 0.44 | 0.39 | 0.39 | 0.68 | 0.68 | 0.65 | 0.63 | 0.60 | 0.62 |
| Temperature 5 cm | 0.75 | 0.74 | 0.76 | 0.75 | 0.73 | 0.77 | 0.97 | 0.97 | 0.97 |
| Temperature 15 cm | 0.82 | 0.82 | 0.83 | 0.80 | 0.80 | 0.81 | 0.98 | 0.97 | 0.98 |
| Temperature 30 cm | 0.92 | 0.91 | 0.93 | 0.83 | 0.81 | 0.86 | 0.97 | 0.98 | 0.97 |
| Temperature 50 cm | 0.93 | 0.93 | 0.93 | 0.90 | 0.89 | 0.91 | 0.95 | 0.95 | 0.94 |
| Temperature 70 cm | 0.96 | 0.96 | 0.96 | 0.93 | 0.93 | 0.93 | 0.96 | 0.97 | 0.96 |