# Peer review of "Physically based approaches incorporating evaporation for early warning predictions of rainfall-induced landslides"

_Natural Hazards and Earth System Sciences, 2017_

## Referee Comment (RC1) · Anonymous Referee #1 · 9 Oct 2017

General comments

I read with interest this research paper investigating the effect of evapotranspiration on physically based models for rainfall-induced landslides. The topic is scientifically significant for the landslide hazard mitigation. I think this paper can be an interesting contribution and is worth to be published but need some major reworking before publication.

First, the introduction is not detailed enough: it lacks of significant contributions in the context of: (1) hillslope hydrology and slope stability and (2) parameters transfer from physical models to real world.

Second, methodology and results should be discussed in more detail specifying some possible limits of the assumptions made. This will lead to more convincing conclusions.

Third, some figures need to be modified, some merged, and some are redundant.

Overall, the paper merges very important aspects of the hillslope hydrology and stability coupling measurements, physical model, and modeling approaches. For this reason I believe it will be suitable for publication and I hope the comments will help the authors to improve the quality and the impact of their manuscript.

Details

To my opinion specific improvements need to cover the following topic:

a) Literature review is limited. In page 2 (line 15 to 20) the authors list a group of physically based hydrological models that neglect evapotranspiration effect. Montgomery and Dietrich, 1994 present a model that uses steady-state hydrology (not suitable for early warning). Moreover, they specify that they use Peff i.e. net rainfall (precipitation less evaporation). Baum et al., 1998 is not the last version of the model and was modified by the Baum et al., 2008 version. It is an event based hydromechanical model, it is not suitable for long term simulation (the report available to: https://pubs.usgs.gov/of/2008/1159/downloads/pdf/OF08-1159.pdf states "TRIGRS is not suitable for modeling long-term effects of alternating periods of rainfall and evapotranspiration, and choosing the correct initial conditions for a given storm is critical to obtaining accurate results"). Formetta et al., 2014 was not correctly cited. It takes into account of evapotranspiration by using the GEOtop model which solves the coupled heat and water balance equations (see Endrizzi et al., 2014). Finally, in the review there is a lack of hydrological models accounting for evapotranspiration (some of them in a simplified way and some of them in a more rigorous way), e.g. Casadei et al. (2003), Šimůnek et al., (2006), Rosso et al., (2006), Ebel et al., (2010), Arnone et al., (2011). I think this is more fair stating both the aspect in the introduction, i. e.: 1) some applications (and models) neglect evapotranspiration because it is considered not the

most relevant process in the analyzed conditions (e.g. Baum et al., 2008; Pagano et al., 2010; Formetta et al., 2016); 2) some applications consider the effect of evapotranspiration with different degree of simplification (Casadei et al. (2003), Rosso et al., (2006), Šimůnek et al., (2006), Ebel et al., (2010), Formetta et al., 2014; Capparelli and Versace (2011); Arnone et al., (2011)) Moreover literature needs to give: i) examples of paper that adopted the same technique of estimating hydrological model parameters in a physical model and use them in real world applications; ii) examples of papers that performed a similar analysis (i.e. evaluation of the effect of evaporation on hillslope hydrology and stability) in other locations or in the same area, stating what make peculiar the current paper (and findings) compared to them.

b) The methodology section should give more emphasis to the novelty presented in this paper. Subsection 2.1 and 2.2 are long description of Rianna et al., 2014a,b; Pagano et al., 2010. It is not clear if the authors are adding something new to that papers: if yes they should point it out more explicitly to facilitate the reader; if not, although it is clear that the background provided by subsections 2.1 and 2.2 is important, authors should consider to summarize them in the main text and detail them in appendix. Same considerations apply to figures 2 to 7: are they showing new data-results compared to Rianna et al., 2014a,b; Pagano et al., 2010?

c) Authors should include in their Discussion and Conclusion considerations concerning the hypothesis used in the paper: i) considering an homogeneous soil whereas many other studies in the area deals with stratified soils; ii) effect of the hysteresis which is evident in the physical model data (Fig. 9-a); iii) transfer in a real world application the same parameters estimated in the physical model (e.g. is there any limit in using the same hydraulic conductivity, how about preferential flow?); iv) the assessment of hillslope stability by a threshold approach neglecting the soil mechanic parameters such as cohesion and friction angle; v) the assumption of one dimensional flow: is the early warning threshold ( estimating neglecting the lateral flow influence) valid for the entire hillslope? Is there any changes in flow behavior at the toe of the hillslope or

in the less steep locations, where lateral flow could be important?

d) The authors should acknowledge explicitly that the analysis presented for the real case application does not use any measured time series of soil suction or soil water content to validate the model.

Specific comments

1) Page 1 line 20: Could you please define "cover" when you use it the first time and use it consistently in the text.

2) Page 2 lines 15 to 20: please consider to update and extend the literature review here.

3) Page 2 line 27 could the Authors please explain which type of model they use.

4) Page 2 line 28: can the Authors please specify in which location those data are collected? Where the landslide happened or in the physical model?

5) Page 3 line 5: could the Authors specify which parameters or at least which type of model parameters they use?;

6) Page 3 line 5: are these procedures new in some theoretical aspect? if yes please specify the novelty, otherwise is better to say "applied" and to reference the procedure applied;

7) Page 3 and 4: please consider to summarize the sections 2.1 and 2.2.

8) Page 3 line 20: can you spell the hydrological variables? Are the data the same used in Pagano et al., 2010?

9) Page 5 section 2.3: Is the model been applied in other similar experiment? If yes, can you cite them?

10) Page 5 line 23: Could the authors please add the units to each variable they use?

11) Page 5 line 24: Could the authors please spell the name and type of the function
12) Page 6 line 2: Could the authors please spell the name and type of the function

13) Page 8 line 6-15: could the authors specify if the procedure has been used for the first time in this paper or could you please reference it?

14) Page 8 line 16: could you please spell the remaining calibrated parameters and the calibration algorithm used? And could you please provide a table of the main parameter values?

15) Page 8 line 19: could the authors please provide a quantification of the agreement in calibration and verification period: for example providing a goodness of fit indices (such as Nash–Sutcliffe, King Gupta Efficiency, Root mean square error, etc); this applies also to soil temperature simulations.

16) Page 9 line 13: Could the authors please motivate the choice of the experimental set up: why 4.5 mm for 60 days? Are those typical value in the study area?

17) Figure 15-c shows that the models tend to behave differently starting from around 10000 KPa. How often the soils experiment those value? Looking at the Figure 9-a the soils had suction values between 1 and 100 KPa and correctly the authors extend the soil water retention curve up to 1000 KPa. However the latter is lower than the 10000 KPa where the models tends to differ (Figure 15-a). Can the Authors comment on this point?

18) Pag 10 line 13: Quantifying the model parameters. Does it mean: using the model parameters estimated thanks to the physical model measurements? Moreover, how the values of the optimal parameter set used in the simulation compares with at-site parameter values used in other studies? Is the order or magnitude the same?

19) The authors should specify the time step of each simulation (physical model and real case both for the input/output variables, and for the inner model time step). In the text (page 5) is it hourly whereas in the figures it seems daily (see captions). If this is true, how this contrasts with the early warning applications? Is there a need of a

sub-daily time step?

20) Please include the NEM model results in Figures 16 and 17 in order to have all the model results in the same figures.

21) Page 12 line 10: Please include in the discussion on the threshold values how it will be influenced by the fact that only one event is considered? How the threshold changes in case of multi-events?

22) Please include some of the limitations of the approaches in the conclusion section and discuss them (see General comments c and d)

23) Figure 10 could be a sub-figure of Figure 9.

24) The paper need to be proof-read possible by a native English speaker. Among them:

- Pag1 line 8: Promptness consider to replace with timeliness;

- Pag1 line 10: Evaporation fluxes consider to replace with evaporative fluxes

- Pag1 line 21: Founding part of their instability: consider to rephrase it

- Pag2 line 1: ranfalls consider to replace with rainfall

- Pag2 line 2-3: Analysys results to triggering cause: rephrase it.

- Pag2 line 17: neglect: remove it

- Pag2 line 19: consider to rephrase as: such an assumption can only be considered reasonable

- Pag2 line 25: arises whether consider to replace with arises as to whether

- Pag2 line 26: The study consider to replace with this study

- Pag3 line 3 and 7: The paper consider to replace with this paper

- Pag3 line 30: obtained consider to replace with used

- Pag4 line 9: between soil consider to replace with between the soil.

- Pag 5 lines 11-13: consider to rephrase it.

- Pag 5 line 25: remove the new paragraph

- Pag 5 line 27: taking into account the possibility of changes consider to replace with taking into account possible changes

- Pag 6 line 6: Remove the

- Pag 6 line 19: cut of: please consider to rephrase it.

- Page 7 line 5: It proves consistent with literature consider to replace with This is consistent with the literature

- Page 7 line 8: dry hot consider to replace with dry and hot

- Page 7 line 10: particularized into please consider to rephrase it

- Page 7 line 12: in the atmosphere temperature consider to replace with in the atmospheric temperature

- Page 9 line 13: remove maintained

- Page 9 line 17: with that water amount consider to replace with with the water amount

- Page 9 line 17: remove that

- Page 9 line 24-25: please rephrase it

- Page 10 line 19: by IEM consider to replace with by the IEM

REFERENCE

Arnone, E., Noto, L. V., Lepore, C., & Bras, R. L. (2011). Physically-based and distributed approach to analyze rainfall-triggered landslides at watershed scale. Geomorphology, 133(3), 121-131.

Baum, R. L., Savage, W. Z., & Godt, J. W. (2008). TRIGRS-A Fortran program for transient rainfall infiltration and grid-based regional slope-stability analysis, version 2.0 (No. 2008-1159). US Geological Survey.

Capparelli, G., & Versace, P. (2011). FLaIR and SUSHI: two mathematical models for early warning of landslides induced by rainfall. Landslides, 8(1), 67-79.

Casadei, M., W. E. Dietrich, and N. L. Miller (2003), Testing a model for predicting the timing and location of shallow landslide initiation in soil- mantled landscapes, Earth Surf. Processes Landforms, 28, 925–950.

Ebel, B. A., Loague, K., & Borja, R. I. (2010). The impacts of hysteresis on variably saturated hydrologic response and slope failure. Environmental Earth Sciences, 61(6), 1215-1225.

Endrizzi, S., Gruber, S., Dall'Amico, M., and Rigon, R.: GEOtop 2.0: simulating the combined energy and water balance at and below the land surface accounting for soil freezing, snow cover and terrain effects, Geosci. Model Dev., 7, 2831-2857, https://doi.org/10.5194/gmd-7-2831-2014, 2014.

Formetta, G., Simoni, S., Godt, J. W., Lu, N., & Rigon, R. (2016). Geomorphological control on variably saturated hillslope hydrology and slope instability. Water Resources Research, 52(6), 4590-4607.

Rosso, R., M. C. Rulli, and G. Vannucchi (2006), A physically based model for the hydrologic control on shallow landsliding, Water Resour. Res., 42, W06410, doi:10.1029/2005WR004369.

Šimůnek, J., Van Genuchten, M. T., & Šejna, M. (2006). The HYDRUS software package for simulating two-and three-dimensional movement of water, heat, and multiple solutes in variably-saturated media. Technical manual, version, 1, 241

---

## Referee Comment (RC2) · Anonymous Referee #2 · 12 Oct 2017

Based on my reviewing, I think this manuscript at least needs some revisions before being accepted for publishing.

The improvements should be addressed as following:

1) In the methodology part, after introduction of the 3 models (BEM, IEM and NEM), an index or a combination of several indexes which will be used for judging the landslide occurring or not should be clearly pointed out in the text, then readers can find these criterions in the following result part and related figures, and have a better understanding the improvement of the BEM, IEM models.

2) More detail discussions should be provided in the result part, especially for the possible limits of the models. As it is stated in the conclusion part, "The models' performance has been assessed by using them to interpret the case history of a landslide and examine their ability to indicate any hydrological peculiarity at the time of the landslide", then arising a question: does the threshold approach in this manuscript is a universal criterion or just feasible in the study area with the soil combination shown in Figure 2? More explaining about the models limitation will make the conclusion more convincing.

3) It would be better if the assessment of slope stability under different models conditions can be provided. How does the suction influence on the slope stability?

4) Suction level or value is an important alarm threshold for landslides induced by rainfall, as these words appear many times in the methodology part, result part, but they are missed in the conclusion part. Conclusion part should include the special important thing which obtained from the study.

5) The units to variables in the equations are missed.

6) Can Figures 16, 17and 18 be shown in one Figure (e.g. 3 model results are shown in one Figure)? Then the difference of 3 models results can be told obviously as well as the novelty of the BEM, IEM models.

7) Figure 3, the sub-title of (e) is missed.

NHESSD

---

## Author Comment (AC1) · 19 Oct 2017

**General comments**

*I read with interest this research paper investigating the effect of evapotranspiration on physically based models for rainfall-induced landslides. The topic is scientifically significant for the landslide hazard mitigation. I think this paper can be an interesting contribution and is worth to be published but need some major reworking before publication.*

Thank you for the effort spent for the Revision and the valuable suggestions.

*First, the introduction is not detailed enough: it lacks of significant contributions in the context of: (1) hillslope hydrology and slope stability*

The text will be modified in order to address the Referee's suggestions also considering proposed literature works.

*and (2) parameters transfer from physical models to real world., methodology and results should be discussed in more detail specifying some possible limits of the assumptions made. This will lead to more convincing conclusions.*

Parameters transfer from physical models to real world represents a key issue in geotechnical problems. Model parameters are typically quantified in laboratory at a scale much smaller than field conditions. In this perspective, according the Authors' view, a strength point of the paper is represented by using for calibration and validation of parameters the findings retrieved by a physical model involving 1 $m^3$ of material forced by realistic boundary conditions provided by actual meteorological evolution, instead of the traditional procedures based on small specimens subject to artificial boundary conditions. In this perspective, a deep comparison among laboratory, lisimeter and field conditions on the same soil involved in the work is reported in Pirone et al. (2016) [doi: 10.1016/j.proeng.2016.08.427] .

In addition, the satisfying performances of the model using so calibrated parameters for interpreting the landslide event is a further indication about the reliability of the whole methodology , worthy to be proposed as a general frame to quantify soil parameters for silty volcanic covers.

*Third, some figures need to be modified, some merged, and some are redundant.*

The proposed modifications will be addressed in the revised paper; specifically a merging is proposed for Figures 9 and 10 and for Figures 16,17 and 18.

*Overall, the paper merges very important aspects of the hillslope hydrology and stability coupling measurements, physical model, and modeling approaches. For this reason I believe it will be suitable for publication and I hope the comments will help the authors to improve the quality and the impact of their manuscript.*

**Details**

To my opinion specific improvements need to cover the following topic:

*a)Literature review is limited.*

Based on suggestions of Referee, more space will be given to reference literature works.

*b) The methodology section should give more emphasis to the novelty presented in this paper.*

We will address the suggestions of the Referee identifying in clearer way the novelty elements presented in the paper; in particular, the idea of i) characterizing the hydrological-thermal evolution in site conditions related to safety conditions as an extrapolation of the behaviour of a reconstituted layer, subject to actual meteorological evolution; ii) adopting the same model for early warning purposes.

*Subsection 2.1 and 2.2 are long description of Rianna et al., 2014a,b; Pagano et al., 2010. It is not clear if the authors are adding something new to that papers: if yes they should point it out more explicitly to facilitate the reader; if not, although it is clear that the background provided by subsections 2.1 and 2.2 is important, authors should consider to summarize the in the main text and detail them in appendix. New*

*Same considerations apply to figures 2 to 7: are they showing new data-results compared to Rianna et al., 2014a,b; Pagano et al., 2010?*

Works by Rianna et al. 2014 a,b report data related to the description of the physical model and to the first two hydrological years (2010-2012). Then, firstly, the paper displays two further years of experimental data concerning water storage, water content and suction, unpublished on other journals. The paper also reports the unpublished whole time history of soil temperatures. Under such clarifications, the Authors would prefer to maintain the consistency of Subsection 2.1 and 2.2 and Figure 2 to 7. The revised version will better specify what data are going to be published for first time.

*c) Authors should include in their Discussion and Conclusion considerations concerning the hypothesis used in the paper:*

> *i) considering an homogeneous soil whereas many other studies in the area deals with stratified soils;*

The Authors will address this point in the "Conclusions" section of the revised version, highlighting how the proposed procedure could be valid only for a homogeneous layer; indeed, only under such assumption 1D conditions can be assumed, as numerically demonstrated by comparing results coming from 1D and 2D analysis (Pagano et al., 2010). However, in this geomorphological context, this condition is widespread resulting also applicable to frequent cases of single homogeneous layers resting on pumices, as the presence of pumices may be replaced by suitable boundary conditions (Reder et al., 2017). Further research works could also extend the procedure to more complex inhomogeneous cases; the eventual assumption of 1D water fluxes has however to be proved, for instance comparing suction predicted in two 2D and 1D hypothesis. At present, studies by Damiano et al. 2017 (10.1016/j.enggeo.2017.02.006), show that one-dimensionality of water fluxes could take place also through sloping layered volcanic soils, so that it is likely that one-dimensionality may be extended at some cases involving layered conditions.

> *ii) effect of the hysteresis which is evident in the physical model data (Fig. 9-a);*

The hysteresis has been neglected in the present study by assuming an unique soil-water characteristic curve fitting all the available observations on calibration time span; the constraints associated to such assumption will be added in the section dealing with parameters calibration; the accuracy loss of the prediction due to neglecting hysteresis is at present topic of new researches, and this will be reported in the "Conclusions" section.

*iii) transfer in a real world application the same parameters estimated in the physical model (e.g. is there any limit in using the same hydraulic conductivity, how about preferential flow?);*

The question raised by the Reviewer about scale problems related to different hydraulic conductivity arising at different scales (from laboratory to field) is challenging and involve all geotechnical problems. Hydraulic conductivity is often measured for small specimens (micro-scale) and then referred to the site (macro-scale) where the scale change may involve different values. It is worth noting however that the "specimen" adopted in the present study is two order of magnitude larger than those typically adopted, and that hydraulic-conductivity is hence measured at a mesoscale, a condition that should make determined values quite close to site ones. Also about such issue, different elements are debated in Pirone et al. (2016) [doi: 10.1016/j.proeng.2016.08.427 ] that will be properly cited in revised version.

*iv) the assessment of hillslope stability by a threshold approach neglecting the soil mechanic parameters such as cohesion and friction angle;*

The chain of events resulting in a landslide of a silty volcanic covers consists in rainfalls, suction drops and induced strength reductions, locally triggering instability due to an internal or external cause, and then propagation of local trigger throughout the cover. The approach followed in the work is aimed to detect the suction levels throughout the slope at which a state predisposing to slope failure is attained. In other words, the philosophy of the approach is that of not dealing with what particular triggering cause able to determine the landslide but, rather, what generalized suction drop determined a slope state prone to propagate a local instability. The suction level at which a predisposing state to landslide takes place depends obviously on strength parameters other than apparent cohesion relating to suction. These are very difficult to characterize and quantify, due to the presence of mechanical effects exerted by root plants. These effects are major, perhaps more significant than other strength contribution, as, in these soils, vegetation is abundant over the entire year. In order to overcome the problems related to characterize vegetation effects and, consequently, set a deterministic slope stability analysis modelling root effects, the approach followed was that to set the early warning prediction straightforwardly on suction levels (or variables relating to suction, as water storage). Taking into consideration that mechanical root effects should in turn be related to suction levels strengthening soils and roots and progressively disappear with suction reductions.

*v) the assumption of one dimensional flow: is the early warning threshold ( estimating neglecting the lateral flow influence) valid for the entire hillslope? Is there any changes in flow behaviour at the toe of the hillslope or in the less steep locations, where lateral flow could be important?*

The answer is in part contained in the discussion to the previous points. In general, the comparison between typical depths of quite homogeneous pyroclastic covers and slope length make reliable for this geomorphological context the assumption of 1D conditions. However, on field, actual conditions may depart from those assumed, (lateral flow influence, fracture increasing flow rate etc.). Local features assumed by the slope hydrology should not affect however that average suction levels throughout the slope making it prone to propagate a local triggering. Generally, local hydrological conditions may be responsible for local triggering, but they are supposed to not affect the state predisposing to propagation.

d) The authors should acknowledge explicitly that the analysis presented for the real case application does not use any measured time series of soil suction or soil water content to validate the model.

This point will be clarified in the presentation of the section treating the discussion of analysis results

*Specific comments*

All specific comments will be taken in great consideration throughout the paper. Among the other ones [item 15], as suggested by the Reviewer, three goodness of fit indices are employed to assess the model's performances (Nash–Sutcliffe, Kling Gupta Efficiency and coefficient of determination); they are discriminated for calibration and validation period. The results reported in revised version provide encouraging performances. [item 16] The value 4.5 mm represents the mean evaporative atmospheric demand estimated through available weather forcing for Summer (JJA) on time span 2010-2014. [item 17] The soil cover usually experiences such values only in the shallower layers during the dry season (see for example Wilson et al., 1994; 1997); nevertheless, the differences between the two approaches are not related in differences in SWCC but, as reported in the text, mainly to reference soil depth from which water is simulated to be extracted according the two interpretative approaches.

---

## Author Comment (AC2) · 19 Oct 2017

*Based on my reviewing, I think this manuscript at least needs some revisions before being accepted for publishing.*

Thank for the effort and the suggestions provided to improve our work.

*The improvements should be addressed as following:*

*1) In the methodology part, after introduction of the 3 models (BEM, IEM and NEM), an index or a combination of several indexes which will be used for judging the landslide occurring or not should be clearly pointed out in the text, then readers can find these criterions in the following result part and related figures, and have a better understanding the improvement of the BEM, IEM models.*

As suggested, in the methodology part, we will explain how (and why) the threshold in terms of suction has been established for judging the landslide occurrence. In this perspective, the approach followed in the work is based on attempts to detect the suction levels throughout the slope at which a state predisposing to propagation of a local trigger propagation takes place. In other words, the philosophy of the approach is that not of dealing with a particular trigger determining the landslide but, rather, of retrieving dynamics regulating generalized suction drop and then a slope state prone to propagate a local instability. The suction level at which a predisposing state to landslide takes place depends obviously on strength parameters other than apparent cohesion relating to suction.

*2) More detail discussions should be provided in the result part, especially for the possible limits of the models. As it is stated in the conclusion part, "The models' performance has been assessed by using them to interpret the case history of a landslide and examine their ability to indicate any hydrological peculiarity at the time of the landslide", then arising a question: does the threshold approach in this manuscript is a universal criterion or just feasible in the study area with the soil combination shown in Figure 2? More explaining about the models limitation will make the conclusion more convincing.*

The study is intended to provide a frame and interpretation tools suitable for all the slopes characterized by similar geomorphological features (area labelled as "Fd" in Picarelli et al.,2008 [ISBN:    978-0-415-41196-7]. To this aim, the back-analysis of the 2005 landslide event is consciously avoided but, rather, the case study is used to test the "predictive capabilities" of the approach. Moreover, threshold suction level to attain slope failure conditions could be different when the single/specific slopes cases are considered. Nevertheless, it is worth noting that on 4[th]  March 2005 also on other slopes characterized by similar features in the area, landslide events of minor relevance occurred while no landslide occurred in the remaining period.

*3) It would be better if the assessment of slope stability under different models conditions can be provided. How does the suction influence on the slope stability?*

The Reviewer's request result quite similar to that formulated by other Anonymous Referee. For these reasons, we report the corresponding answer

The chain of events resulting in a landslide of a silty volcanic covers consists in rainfalls, suction drops and induced strength reductions, locally triggering instability due to an internal or external cause, and then propagation of local trigger throughout the cover. The approach followed in the work is aimed to detect the suction levels throughout the slope at which a state predisposing to slope failure is attained. In other words, the philosophy of the approach is that of not dealing with what particular triggering cause able to determine the landslide but, rather, what generalized suction drop determined a slope state prone to propagate a local instability. The suction level at which a predisposing state to landslide takes place depends obviously on strength parameters other than apparent cohesion relating to suction. These are very difficult to characterize

and quantify, due to the presence of mechanical effects exerted by root plants. These effects are major, perhaps more significant than other strength contribution, as, in these soils, vegetation is abundant over the entire year. In order to overcome the problems related to characterize vegetation effects and, consequently, set a deterministic slope stability analysis modelling root effects, the approach followed was that to set the early warning prediction straightforwardly on suction levels (or variables relating to suction, as water storage). Taking into consideration that mechanical root effects should in turn be related to suction levels strengthening soils and roots and progressively disappear with suction reductions.

*4) Suction level or value is an important alarm threshold for landslides induced by rainfall, as these words appear many times in the methodology part, result part, but they are missed in the conclusion part. Conclusion part should include the special important thing which obtained from the study.*

Such issue will be stressed in the revised version of the paper. Thank you for the suggestion.

5) The units to variables in the equations are missed.

We will introduce in the revised version the units to variables cited in the Manuscript

6) Can Figures 16, 17and 18 be shown in one Figure (e.g. 3 model results are shown in one Figure)? Then the difference of 3 models results can be told obviously as well as the novelty of the BEM, IEM models.

As suggested we have merged Figures 16, 17 and 18 in a single Figure

7) Figure 3, the sub-title of (e) is missed.

We will add the sub-title of (e) for Figure 3

---

## Referee Report (RR1)

[referee-annotated manuscript omitted]

---

## Referee Report (RR2)

**Anonymous Referee #1 Received and published: 9 October 2017**

Dear authors, below you can find the revision of the paper. The reviewer questions posted in the first round and the authors' answers to them are in normal font. The new questions/suggestions for the second round are in bold.

**General comments**

I read with interest this research paper investigating the effect of evapotranspiration on physically based models for rainfall-induced landslides. The topic is scientifically significant for the landslide hazard mitigation. I think this paper can be an interesting contribution and is worth to be published but need some major reworking before publication.

Dear Referee, thank you for the effort spent for the Revision and the valuable suggestions. We have addressed your suggestions whenever possible. You can find the modified parts in the revised text in red. For details comment, pages and lines where text has been modified to accomplish your suggestions are specifically indicated.

First, the introduction is not detailed enough: it lacks of significant contributions in the context of: (1) hillslope hydrology and slope stability and (2) parameters transfer from physical models to real world. Regarding point (1), we have revised the Introduction considering references to work dealing with hillslope hydrology and slope stability for rainfall-induced landslides in Campania Region and other geomorphological contexts. Regarding point (2), parameters transfer from physical models to real world represents a key issue in geotechnical problems. Model parameters are typically quantified in laboratory at a scale much smaller than field conditions. In this perspective, according the Authors' view, a strength point of the paper is represented by using for calibration and validation of parameters the findings retrieved by a physical model involving 1 m3 of material forced by realistic boundary conditions provided by actual meteorological evolution, instead of the traditional procedures based on small specimen subject to artificial boundary conditions. In this perspective, a deep comparison among laboratory, lysimeter and field conditions on the same soil involved in the work is reported in Pirone et al. (2016) [doi: 10.1016/j.proeng.2016.08.427]. In addition, the satisfying performances of the model using so calibrated parameters for interpreting the landslide event is a further indication about the reliability of the whole methodology, worthy to be proposed as a general frame to quantify soil parameters for silty volcanic covers.

**Regarding point (2), the authors should add in the introduction section of the paper the consideration presented in the answer, where they**

**express the novelty of the paper:** "Model parameters are typically quantified in laboratory at a scale much smaller than field conditions. In this perspective, according the Authors' view, a strength point of the paper is represented by using for calibration and validation of parameters the findings retrieved by a physical model involving 1 m3 of material forced by realistic boundary conditions provided by actual meteorological evolution, instead of the traditional procedures based on small specimen subject to artificial boundary conditions. In this perspective, a deep comparison among laboratory, lysimeter and field conditions on the same soil involved in the work is reported in Pirone et al. (2016) [doi:

10.1016/j.proeng.2016.08.427]. In addition, the satisfying performances of the model using so calibrated parameters for interpreting the landslide event is a further indication about the reliability of the whole methodology, worthy to be proposed as a general frame to quantify soil parameters for silty volcanic covers"

**Moreover, the authors should better specify what makes this work/paper different from Pirone et al. 2016 in the Introduction section as well.**

Second, methodology and results should be discussed in more detail specifying some possible limits of the assumptions made. This will lead to more convincing conclusions. Assumptions and constraints have been reported in the manuscript revised version when methodology and results are discussed and have been summarized in Conclusions. Third, some figures need to be modified, some merged, and some are redundant.

The proposed modifications have been addressed in the revised paper; specifically a merging is proposed for Figures 9 and 10 and for Figures 16,17 and 18. Overall, the paper merges very important aspects of the hillslope hydrology and stability coupling measurements, physical model, and modeling approaches. For this reason I believe it will be suitable for publication and I hope the comments will help the authors to improve the quality and the impact of their manuscript.

**Details**

To my opinion specific improvements need to cover the following topic: a) Literature review is limited. In page 2 (line 15 to 20) the authors list a group of physically based hydrological models that neglect evapotranspiration effect. Montgomery and Dietrich, 1994 present a model that uses steady-state hydrology (not suitable for early warning). Moreover, they specify that they use Peff i.e. net rainfall (precipitation less evaporation). Baum et al., 1998 is not the last version of the model and was modified by the Baum et al., 2008 version. It is an event based hydromechanical model, it is not suitable for long term simulation (the report available to:

https://pubs.usgs.gov/of/2008/1159/downloads/pdf/OF08-1159.pdf states "TRIGRS is not suitable for modeling long-term effects of alternating periods of rainfall and evapotranspiration, and choosing the correct initial conditions for a given storm is critical to obtaining accurate results"). Formetta et al., 2014 was not correctly cited. It takes into account of evapotranspiration by using the GEOtop model which solves the coupled heat and water balance equations (see Endrizzi et al., 2014). Finally, in the review there is a lack of hydrological models accounting for evapotranspiration (some of them in a simplified way and some of them in a more rigorous way), e.g. Casadei et al. (2003), Šim°unek et al., (2006), Rosso et al., (2006), Ebel et al., (2010), Arnone et al., (2011). I think this is more fair stating both the aspect in the introduction, i. e.: 1) some applications (and models) neglect evapotranspiration because it is considered not the most relevant process in the analyzed conditions (e.g. Baum et al., 2008; Pagano et al., 2010; Formetta et al., 2016); 2) some applications consider the effect of evapotranspiration with different degree of simplification (Casadei et al. (2003), Rosso et al., (2006), Šim°unek et al., (2006), Ebel et al., (2010), Formetta et al., 2014; Capparelli and Versace (2011); Arnone et al., (2011)) Moreover literature needs to give: i) examples of paper that adopted the same technique of estimating hydrological model parameters in a physical model and use them in real world applications; ii) examples of papers that performed a similar analysis (i.e. evaluation of the effect of evaporation on hillslope hydrology and stability) in other locations or in the same area, stating what make peculiar the current paper (and findings) compared to them. Based on suggestions of Referee, more space will be given to reference literature works including both references suggested by Referee and others referring to Campania pyroclastic covers (Page 2 line 16-19; Page 2 line 23-31).

**The new sentence added in the revised paper states:** "This implies that in several applications evaporation is neglected at all (e.g., Baum et al., 2008; Pagano et al., 2010; Formetta et al., 2016) or taken in to account following simplified approaches (Casadei et al., 2003; Rosso et al., 2006; Šimunek et al., 2006; Ebel et al., 2010; Formetta et al., 2014; Capparelli 25 and Versace, 2011; Arnone et al., 2011). Complete approaches, modelling internal and boundary evaporation basing on hydrothermal approaches, were taken into account in studies referred to slopes in fine-grained soils differing substantially from those involved in the case in hand (Cui et al., 2005; An et al., 2017; Song et al., 2016)."

**Consider to rephrase:** "This implies that in several applications evaporation is neglected at all".

It seems that the cited studies neglect evapotranspiration for sake of simplicity but is not the case. In each paper the explanation is given of the reason of neglecting it (in two of them for example it has been assumed that in an intense rainfall event triggering a landslide evapotranspiration had not a high impact as the rainfall intensity). I invite the author to specify the reason for which evapotranspiration has been neglected: it was considered less important than rainfall intensity during a highly intense precipitation event that triggered landslides. Consider to rephrase: "or taken in to account following simplified approaches".

The authors merge examples that consider evapotranspiration in a simplified way (like precipitation less evapotranspiration (see Rosso et al., 2006)) with example in which it is taken into account in a more complex way, dynamically solving the coupled energy and water budget (see Šimunek et al., 2006; Formetta et al., 2014). It would be fair to state this difference among the approaches. From the sentence it seems that solving the couple water-energy budget is a simplified approach to account for evapotranspiration.

b) The methodology section should give more emphasis to the novelty presented in this paper. Subsection 2.1 and 2.2 are long description of Rianna et al., 2014a,b; Pagano et al., 2010. It is not clear if the authors are adding something new to that papers: if yes they should point it out more explicitly to facilitate the reader; if not, although it is clear that the background provided by subsections 2.1 and 2.2 is important, authors should consider to summarize them in the main text and detail them in appendix. Same considerations apply to figures 2 to 7: are they showing new data-results compared to Rianna et al., 2014a,b; Pagano et al., 2010?

We have addressed the suggestions of the Referee identifying in clearer way the novelty elements presented in the paper; in particular, the idea of i) characterizing the hydrological-thermal evolution in site conditions related to safety conditions as an extrapolation of the behavior of a reconstituted layer, subject to actual meteorological evolution; ii) adopting the same model for early warning purposes. (Page 3 line 16-17)

Moreover, works by Rianna et al. (2014 a,b) report data related to the description of the physical model and to the first two hydrological years (2010-2012). Then, firstly, the paper displays two further years of experimental data concerning water storage, water content and suction, unpublished on other journals. The paper also reports the unpublished evolution of soil temperatures. Under such clarifications, the Authors would prefer to maintain the consistency of Subsection 2.1 and 2.2 and Figure 2 to 7. The revised version better specifies what data are going to be published for first time.

c) Authors should include in their Discussion and Conclusion considerations concerning the hypothesis used in the paper:

i) considering an homogeneous soil whereas many other studies in the area deals with stratified soils;

We have addressed this point in the revised version of the paper, highlighting how the proposed procedure could be valid only for a homogeneous layer; indeed, only under such assumption 1D conditions can be assumed, as numerically demonstrated by comparing results coming from 1D and 2D analysis (Pagano et al., 2010). However, in this geomorphological context, this condition is widespread resulting also applicable to frequent cases of single homogeneous layers resting on pumices, as the presence of pumices may be replaced by suitable boundary conditions (Reder et al., 2017). Further research works could also extend the procedure to more complex inhomogeneous cases; the eventual assumption of 1D water fluxes has however to be proved, for instance comparing suction predicted in two 2D and 1D hypothesis. At present, studies by Damiano et al. 2017, show that one-dimensionality of water fluxes could take place also through sloping layered volcanic soils, so that it is likely that one-dimensionality may be extended at some cases involving layered conditions. (Page 5 line 15-20).

**The authors should consider to add and cite existing studies that assumes stratified soils in their simulations in the same study area.**

ii) effect of the hysteresis which is evident in the physical model data (Fig. 9a);

The hysteresis has been neglected in the present study by assuming a unique soil-water characteristic curve fitting all the available observations on calibration time span; the accuracy loss of the prediction due to neglecting hysteresis is at present topic of new researches (Page 10 line 3-5; Page 15 line 1-3).

Please, consider to explain better the concept on the paper, adding in the paper what it is stated in the answer to the reviewer i.e.: merging measured data at different depth to obtain a unique swrc (The hysteresis has been neglected in the present study by assuming a unique soil-water characteristic curve fitting all the available observations on calibration time span; the accuracy loss of the prediction due to neglecting hysteresis is at present topic of new researches). **Please, consider to specify the reason of this assumption and what are the limitations of this approach.**

iii) transfer in a real world application the same parameters estimated in the physical model (e.g. is there any limit in using the same hydraulic conductivity, how about preferential flow?);

The question raised by the Reviewer about scale problems related to different hydraulic conductivity arising at different scales (from laboratory to field) is challenging and involve all geotechnical problems. Hydraulic conductivity is often measured for small specimens (micro-scale) and then referred to the site (macro-scale) where the scale change may involve different values. It is worth noting however that the "specimen" adopted in the present study is two order of magnitude larger than those typically adopted, and that hydraulic-conductivity is hence measured at a mesoscale, a condition that should make determined values quite close to site ones. Also about such issue, different elements are debated in Pirone et al. (2016) that will be properly cited in revised version (Page 11 line 22-25).

**Please consider to add in the text the specimen size and the positive effects they have according the authors point of view.**

iv) the assessment of hillslope stability by a threshold approach neglecting the soil mechanic parameters such as cohesion and friction angle;

The chain of events resulting in a landslide of a silty volcanic covers consists in rainfalls, suction drops and induced strength reductions, locally triggering instability due to an internal or external cause, and then propagation of local trigger throughout the cover. The approach followed in the work is aimed to detect the suction levels throughout the slope at which a state predisposing to slope failure is attained. In other words, the philosophy of the approach is that of not dealing with what particular triggering cause able to determine the landslide but, rather, what generalized suction drop determined a slope state prone to propagate a local instability. The suction level at which a predisposing state to landslide takes place depends obviously on strength parameters other than apparent cohesion relating to suction. These are very difficult to characterize and quantify, due to the presence of mechanical effects exerted by root plants. These effects are major, perhaps more significant than other strength contribution, as, in these soils, vegetation is abundant over the entire year. In order to overcome the problems related to characterize vegetation effects and, consequently, set a deterministic slope stability analysis modelling root effects, the approach followed was that to set the early warning prediction straightforwardly on suction levels (or variables relating to suction, as water storage). Taking into consideration that mechanical root effects should in turn be related to suction levels strengthening soils and roots and progressively disappear with suction reductions (Page 11 line 26-33; Page 12 line 1-4).

**I encourage the authors to cite works that compute the slope instability using this approach. Moreover, it would be important to discuss about the possibility to extend the threshold to near locations?**

v) the assumption of one dimensional flow: is the early warning threshold (estimating neglecting the lateral flow influence) valid for the entire hillslope? Is there any changes in flow behavior at the toe of the hillslope or in the less steep locations, where lateral flow could be important? The answer is in part contained in the discussion to the previous points. In general, the comparison between typical depths of quite homogeneous pyroclastic covers and slope length make reliable for this geomorphological context the assumption of 1D conditions. However, on field, actual conditions may depart from those assumed, (lateral flow influence, fracture increasing flow rate etc.). Local features assumed by the slope hydrology should not affect however that average suction levels throughout the slope making it prone to propagate a local triggering. Generally, local hydrological conditions may be responsible for local triggering, but they are supposed to not affect the state predisposing to propagation (Page 5 line 15-20).

**Could you please consider to add in the paper the discussion provided**

**in this answer** (In general, the comparison between typical depths of quite homogeneous pyroclastic covers and slope length make reliable for this geomorphological context the assumption of 1D conditions. However, on field, actual conditions may depart from those assumed, (lateral flow influence, fracture increasing flow rate etc.). Local features assumed by the slope hydrology should not affect however that average suction levels throughout the slope making it prone to propagate a local triggering. Generally, local hydrological conditions may be responsible for local triggering, but they are supposed to not affect the state predisposing to propagation) with proper references supporting each important statement?

d) The authors should acknowledge explicitly that the analysis presented for the real case application does not use any measured time series of soil suction or soil water content to validate the model.

This point has been clarified in the presentation of the section treating the discussion of analysis results (Page 15 line 18-20).

Probably this could be moved when the methodology is described and also in the abstract.

**Specific comments**

1) Page 1 line 20: Could you please define "cover" when you use it the first time and use it consistently in the text.

We have modified the manuscript following Reviewer indication (Page 1 line 20-23).

2) Page 2 lines 15 to 20: please consider to update and extend the literature review here. Literature review has been updated and extended considering the Reference suggested by Referee (Page 2 line 16-19; Page 2 line 23-31).
3) Page 2 line 27 could the Authors please explain which type of model they use.

We have specified which type of model we have adopted (Page 3 line 2).

**Are the authors meaning a "hydrological" model?**

4) Page 2 line 28: can the Authors please specify in which location those data are collected? Where the landslide happened or in the physical model? We have indicated in which location data are collected; specifically, data are obtained from a weather station located where the landslide happened (Page 3 line 6-8).

5) Page 3 line 5: could the Authors specify which parameters or at least which type of model parameters they use?

We have specified which models we adopt; the models are introduced as suggested by Reviewer in (item 3) so for this comment we have only slightly modified the text (Page 3 line 12).

**Reading the sentence the only visible modification in the sentence is the following in red:**

Since these three cited models are presumed to be operating in real time, namely receiving recorded meteorological variables as input data and returning variables relating to slope safety conditions as output data, they need to be applied to simplified geometrical and mechanical patterns to save as much analysis time as possible.

**To which patterns the authors refers? Can they please make some examples?**

6) Page 3 line 5: are these procedures new in some theoretical aspect? if yes please specify the novelty, otherwise is better to say "applied" and to reference the procedure applied;

The procedures used for the work in hand are new in some theoretical aspect; we have specified this issue (Page 3 line 16-17).

7) Page 3 and 4: please consider to summarize the sections 2.1 and 2.2.
Works by Rianna et al. (2014a,b) show data related to the physical model description and first two hydrological years (2010-2012). Then, firstly, the paper displays two further years of experimental data concerning water storage, water content and suction, unpublished on other journals. The paper also reports the unpublished development of time of soil temperatures. Under such clarifications, the Authors would prefer to maintain the consistency of Subsection 2.1 and 2.2 and Figure 2 to 7. The revised version will better specify what data are going to be published for first time. (Page 4 line 28-29).
8) Page 3 line 20: can you spell the hydrological variables? Are the data the same used in Pagano et al., 2010?

We have indicated the hydrological variables reported in Figure 3; we have specified that only precipitation is reported and used in Pagano et al. (2010) (Page 3 line 31-32 Page 4 line 1)

9) Page 5 section 2.3: Is the model been applied in other similar experiment?If yes, can you cite them?

Details about previous applications of such model are reported in the revised manuscript (Page 7 line 19-21).

10) Page 5 line 23: Could the authors please add the units to each variable they use?

Units have been added for all the variables used (Page 6 line 2-5, line 11-12, line 19-19 Page 7 line 10-11).

11) Page 5 line 24: Could the authors please spell the name and type of the function

Ssoil water characteristic curve (SWCC) and hydraulic conductivity function (HCF) have been obtained using the Van Genuchten (1980) and Mualem-Van Genuchten (Mualem, 1976) equations. The model parameters for both functions have been summarized in Table 1.

12) Page 6 line 2: Could the authors please spell the name and type of the function

Thermal functions have been obtained using *ad-hoc* exponential equations whose parameters have been summarized in Table 1.

**Please add the units where missing.**

13) Page 8 line 6-15: could the authors specify if the procedure has been used for the first time in this paper or could you please reference it?The procedure used to obtain soil parameters involves for some variables novelty in interpretation stages. This is the case for example of Ch. This point has been reported in the revised manuscript (Page 8 line 26).

14) Page 8 line 16: could you please spell the remaining calibrated parameters and the calibration algorithm used? And could you please provide a table of the main parameter values?

We have specified the remaining calibrated function (HCF) and provided the parameter values in Table 1 (Page 9 line 5).

15) Page 8 line 19: could the authors please provide a quantification of the agreement in calibration and verification period: for example providing a goodness of fit indices (such as Nash–Sutcliffe, Kling Gupta Efficiency, Root mean square error, etc); this applies also to soil temperature simulations. As suggested by the Reviewer, three goodness of fit indices are employed to assess the model's performances (Nash–Sutcliffe, Kling Gupta Efficiency and coefficient of determination); they are discriminated for calibration and

validation period. Results are reported in Table 2. The findings result to be very encouraging. Brief details about indices and results will be added in revised version (Page 9 line 29-32 Page 10 line 115).

**Could you please consider to revise the table using Kling-Gupta, and Suction 15 cm, etc (i.e. trying to spell the names and captions)**

16) Page 9 line 13: Could the authors please motivate the choice of the experimental set up: why 4.5 mm for 60 days? Are those typical value in the study area?

The value 4.5 mm represents the mean evaporative atmospheric demand estimated through available weather forcing for Summer (JJA) on time span 2010-2014 (Page 10 line 13-14).

17) Figure 15-c shows that the models tend to behave differently starting from around 10000 KPa. How often the soils experiment those value? Looking at the Figure 9-a the soils had suction values between 1 and 100 KPa and correctly the authors extend the soil water retention curve up to 1000 KPa. However the latter is lower than the 10000 KPa where the models tends to differ (Figure 15-a). Can the Authors comment on this point? The soil cover usually experiences such values only in the shallower layers during the dry season (see for example Wilson et al., 1994; 1997); nevertheless, the differences between the two approaches are not related in differences in SWCC but, as reported in the text, mainly to reference soil depth from which water is simulated to be extracted according the two interpretative approaches. (Page 11 line 10-14). We have also added a comment for SWCC (Page 8 line 15-17) and modified Figure 9(a) extrapolating data up to 10000kPa.

18) Pag 10 line 13: Quantifying the model parameters. Does it mean: using the model parameters estimated thanks to the physical model measurements? Moreover, how the values of the optimal parameter set used in the simulation compares with at-site parameter values used in other studies? Is the order or magnitude the same?

We have entered a new section (3.2.1 Preliminary assumptions and considerations) in which, among the others, this item has been specified referring to Pirone et al. (2016). Such work compares laboratory, lysimeter

and field monitoring for the soil in hand showing how the lysimeter represents an excellent tool for assessing soil hydraulic properties. (Page 11 line 23-25). 19) The authors should specify the time step of each simulation (physical model and real case both for the input/output variables, and for the inner model time step). In the text (page 5) is it hourly whereas in the figures it seems daily (see captions). If this is true, how this contrasts with the early warning applications? Is there a need of a sub-daily time step? We have specified time step of each simulation for input/output and inner model time step (Page 8 line 5-6).

20) Please include the NEM model results in Figures 16 and 17 in order to have all the model results in the same figures.

As suggested also by Reviewer 2, we have merged Figure 15, 16 and 17 in one Figure (see Figure 15).

21) Page 12 line 10: Please include in the discussion on the threshold values how it will be influenced by the fact that only one event is considered? How the threshold changes in case of multi-events?

We have included a discussion on threshold value focusing on the influence that it is based only on one event and how the threshold could change for multi-events (Page 15 line 8-17).

22) Please include some of the limitations of the approaches in the conclusion section and discuss them (see General comments c and d)

Limitation of the approaches with discussion has been included (Page 14 line 24-30 Page 15 line 1-6).

23) Figure 10 could be a sub-figure of Figure 9.

We have merged Figure 10 with Figure 9. Figure 10 is now Figure 9c.

24) The paper need to be proof-read possible by a native English speaker. Among them:

- Pag1 line 8: Promptness consider to replace with timeliness;

- Pag1 line 10: Evaporation fluxes consider to replace with evaporative fluxes

- Pag1 line 21: Founding part of their instability: consider to rephrase it
- Pag2 line 1: ranfalls consider to replace with rainfall
- Pag2 line 2-3: Analysys results to triggering cause: rephrase it.
- Pag2 line 17: neglect: remove it
- Pag2 line 19: consider to rephrase as: such an assumption can only be

considered reasonable

- Pag2 line 25: arises whether consider to replace with arises as to whether
- Pag2 line 26: The study consider to replace with this study
- Pag3 line 30: obtained consider to replace with used
- Pag4 line 9: between soil consider to replace with between the soil.
- Pag 5 lines 11-13: consider to rephrase it.
- Pag 5 line 25: remove the new paragraph
- Pag 5 line 27: taking into account the possibility of changes consider to replace with taking into account possible changes
- Pag 6 line 6: Remove the
- Pag 6 line 19: cut of: please consider to rephrase it.
- Page 7 line 5: It proves consistent with literature consider to replace with This is consistent with the literature
- Page 7 line 8: dry hot consider to replace with dry and hot
- Page 7 line 10: particularized into please consider to rephrase it
- Page 7 line 12: in the atmosphere temperature consider to replace with in the atmospheric
- temperature
- Page 9 line 13: remove maintained
- Page 9 line 17: with that water amount consider to replace with with the water amount
- Page 9 line 17: remove that
- Page 9 line 24-25: please rephrase it
- Page 10 line 19: by IEM consider to replace with by the IEM

All the points have been accomplished.

**REFERENCE**

Arnone, E., Noto, L. V., Lepore, C., & Bras, R. L. (2011). Physically-based and distributed approach to analyze rainfall-triggered landslides at watershed scale. Geomorphology, 133(3), 121-131.

Baum, R. L., Savage, W. Z., & Godt, J. W. (2008). TRIGRS-A Fortran program for transient rainfall infiltration and grid-based regional slope-stability analysis, version 2.0 (No. 2008-1159). US Geological Survey.

Capparelli, G., & Versace, P. (2011). FLaIR and SUSHI: two mathematical models for early warning of landslides induced by rainfall. Landslides, 8(1),

**67-79.**

Casadei, M., W. E. Dietrich, and N. L. Miller (2003), Testing a model for predicting the timing and location of shallow landslide initiation in soil- mantled landscapes, Earth Surf. Processes Landforms, 28, 925–950.

Ebel, B. A., Loague, K., & Borja, R. I. (2010). The impacts of hysteresis on variably saturated hydrologic response and slope failure. Environmental Earth Sciences, 61(6), 1215-1225.

Endrizzi, S., Gruber, S., Dall'Amico, M., and Rigon, R.: GEOtop 2.0: simulating the combined energy and water balance at and below the land surface accounting for soil freezing, snow cover and terrain effects, Geosci. Model Dev., 7, 2831-2857,https://doi.org/10.5194/gmd-7-2831-2014, 2014. Formetta, G., Simoni, S., Godt, J. W., Lu, N., & Rigon, R. (2016). Geomorphological control on variably saturated hillslope hydrology and slope instability. Water Resources Research, 52(6), 4590-4607. Rosso, R., M. C. Rulli, and G. Vannucchi (2006), A physically based model for the hydrologic control on shallow landsliding, Water Resour. Res., 42, W06410,doi:10.1029/2005WR004369.

Šim°unek, J., Van Genuchten, M. T., & Šejna, M. (2006). The HYDRUS software package for simulating two-and three-dimensional movement of water, heat, and multiple solutes in variably-saturated media. Technical manual, version, 1, 241

---

## Author Response (AR3)

**ANNOTATIONS:**

All the modifications and improvements required by Editor have been implemented. Moreover, it worth reporting that in "2.1 Field experimental data: the Nocera Inferiore 2005 landslide" the highlighted point is introduced in Pag. 4 Line 19-20 and not in 15-16 as wrongly reported in our previous revision.
All the required modifications are reported in red while the suggestions for text improvements in green.

[revised manuscript text omitted]